# Dissolved carbon flow to particulate organic carbon enhances soil carbon sequestration

Qintana Si[1], Kangli Chen[1], Bin Wei[1], Yaowen Zhang[1], Xun Sun[1], Junyi Liang[1]

[1]Department of Grassland Resource and Ecology, College of Grassland Science and Technology, China Agricultural University, Beijing 100193, China

*Correspondence to*: Junyi Liang (+86 10 62733381; liangjunyi@cau.edu.cn)

**Abstract.** Particulate organic carbon (POC) and mineral-associated organic carbon (MAOC), which are two primary components of the soil carbon (C) reservoir, have different physical and chemical properties and biochemical turnover rates. Microbial necromass entombment is a primary mechanism for MAOC formation from fast-decaying plant substrates, whereas POC is typically considered the product of structural litter via physical fragmentation. However, emerging evidence shows that microbial by-products derived from labile C substrates can enter the POC pool. To date, it is still unclear to what extent dissolved C can enter the POC pool and how it affects the subsequent long-term SOC storage. Our study here, through a [13]C-labeling experiment in 10 soils from 5 grassland sites as well as a modeling analysis, showed that up to 12.29% of isotope-labeled glucose-C (i.e., dissolved C) was detected in the POC pool. In addition, the glucose-derived POC was correlated with [13]C-MBC and the fraction of clay and silt, suggesting that the flow of dissolved C to POC is dependent on interactions between soil physical and microbial processes. The modeling analysis showed that ignoring the C flow from MBC to POC significantly underestimated soil C sequestration by up to 53.52% across the 10 soils. The results emphasize that the soil mineral-regulated microbial process, besides the plant structural residues, is a significant contributor to POC, acting as a vital component in SOC dynamics.

**Keywords:** soil carbon formation, soil carbon sequestration, soil carbon modeling, particulate organic carbon, dissolved organic carbon, soil carbon input, glucose, grassland, fencing

# 1 Introduction

As the largest terrestrial carbon (C) pool, soil organic C (SOC) plays a vital role in regulating global climate change through C emissions and sequestration (White et al., 2000; Chapin et al., 2011; Wiesmeier et al., 2019; Basile-Doelsch et al., 2020; Bai and Cotrufo, 2022). Carbon from living roots can be stabilized in the form of mineral-associated organic C (MAOC), while plant residues can enter the soil as particulate organic C (POC), which has different features from MAOC (Cotrufo et al., 2019; Sokol et al., 2019b; Lavallee et al., 2020). MAOC is generally small molecular organo-mineral complexes with relatively low

C/nitrogen (N) ratios (C/N ratios). Being associated with soil minerals and occluded in the silt- and clay-sized aggregates, MAOC has a longer mean residence time than POC and thus is considered the key to long-term soil sequestration (Baldock and Skjemstad, 2000). In contrast, POC is usually considered the product of physically fragmented structural residues and is more susceptible to external environmental changes (Benbi et al., 2014; Lugato et al., 2021). Although physically dividing SOC into POC and MAOC is relatively easy, the microbe-mediated SOC dynamics is a continuous process, and it is difficult

to separate its biochemical and physical processes completely (Lehmann et al., 2020). During the gradual decomposition of plant residues by microorganisms, the surface of fresh litter combines with soil minerals in the presence of microbial products (which act as binders in the soil), forming heavy-POC (or coarse-MAOC, >53 μm and >1.6 – 1.85 g cm$^{-3}$) (Samson et al., 2020). The plant-soil interface in heavy-POC promotes the formation of soil aggregates directly, becoming the hotspot for SOC formation (Witzgall et al., 2021). Meanwhile, given that heavy-POC is a combination of plant residues, microbial

products, and soil minerals, and that heavy-POC has a similar C:N ratio to MAOC, it is reasonable to hypothesize that the fragmentation of heavy-POC promotes the formation of MAOC and is the precursor for MAOC (Samson et al., 2020; Zhang et al., 2021).

Dissolved C input from living roots (i.e. rhizodeposits), which has a dominant effect on the net formation of SOC, is considered approximately 2 to 13 times more efficient than litter inputs in forming SOC (Sokol et al., 2019b). The Microbial Efficiency-

Matrix Stabilization (MEMS) framework also suggests that labile plant C inputs are a major source of microbial products, which are more efficiently utilized by microorganisms than recalcitrant ones (Cotrufo et al., 2013). However, the labile C input also plays a critical role in destabilizing SOC as well (Kuzyakov et al., 2000; Keiluweit et al., 2015). Most of the literature emphasizes that labile plant substrates with low molecular weight– such as glucose and other dissolved C – are primary sources of MAOC through physical absorption and microbial *in vivo* turnover via cell uptake-biosynthesis-growth-death (Bai and

Cotrufo, 2022; Mikutta et al., 2019; Sokol et al., 2019a; Liang et al., 2017). However, the potential for microbial products derived from labile C to stick to semi-decomposed plant residues and connect with minerals to become POC has received much less attention.

As an important component of SOC, POC is pivotal in predicting SOC sequestration as well. A few mechanistic models propose POC formation from microbial metabolism, but there is a limited understanding of the factors controlling POC

formation (Li et al., 2014; Robertson et al., 2019; Cotrufo and Lavallee, 2022). Specifically, direct evidence is still lacking to

what extent dissolved substrate (e.g., glucose) contributes to POC. Additionally, how the dissolved substrates-originated POC affects SOC formation is rarely studied.

Meanwhile, the soil C dynamics are sensitive to land use changes (Del Galdo et al., 2003; Grandy and Robertson, 2007). Overgrazing and conversion of grasslands to farmlands have resulted in significant ecosystem degradation in the grasslands of
northern China (Wang et al., 2023; Buisson et al., 2022). Fencing is a widely used strategy in order to retard and reverse the grassland degradation. To date, it has been well-studied that fencing can improve the plant community structure of degraded grasslands, increase species diversity, improve soil structure, promote soil microbial biomass and enzyme activity (Lu et al., 2018; Bardgett et al., 2021). However, how differently dissolved substrates affect POC and MAOC dynamics in fenced and grazed grasslands is still unclear.

In this study, we first collected soil samples from fenced and grazed grasslands from 5 sites (Table 1). Then, we conducted an incubation experiment by adding [13]C-labeled glucose solution to the 10 soils. At the end of the experiment, glucose-derived [13]C in dissolved organic C (DOC), microbial biomass C (MBC), POC, and MAOC were assessed. Then, we conducted a modeling experiment to simulate SOC dynamics at different C addition scenarios with and without a dissolved C flow from MBC to POC. This study was to answer the following three questions: (i) to what extent the added glucose contributes to POC?
(ii) what factors control the dissolved C flow to POC in the fenced and grazed grasslands across sites? (iii) how does dissolved substrates-originated POC affect SOC sequestration? To answer the questions, we had three hypotheses. First, dissolved C can get into the POC pool in addition to the MAOC pool due to interactions between soil physical and biochemical processes. Second, the rate of POC conversion from glucose is dependent upon microbial activity due to the land use change across sites. Finally, neglecting the process of dissolved C flow to POC leads to an underestimation of SOC sequestration.

**Table 1: Information of the sampling sites and soil physical and chemical properties (mean ± standard error).**

| Site | Fencing treatment | Abbrev-iation | Longitude (°E) | Latitude (°N) | Altitude (m) | Mean annual precipitation (mm) | Mean annual temperature (°C) | Clay (%) | Silt (%) | Sand (%) | pH | SOC (g kg$^{-1}$) |
|---|---|---|---|---|---|---|---|---|---|---|---|---|
| DL | fencing | DL$_{fencing}$ | 116.27 | 42.06 | 1306.22 | 378.00 | 3.30 | 9.4 | 17.7 | 71.3 | 7.52±0.07 | 57.08±3.53 |
| DL | grazing | DL$_{grazing}$ | | | | | | 10.3 | 18.3 | 67.2 | 7.27±0.20 | 59.70±1.62 |
| GY | fencing | GY$_{fencing}$ | 115.59 | 41.78 | 1391.95 | 398.40 | -1.40 | 9.2 | 20.1 | 70.6 | 8.02±0.21 | 40.13±2.67 |
| GY | grazing | GY$_{grazing}$ | | | | | | 10.0 | 14.9 | 74.1 | 7.57±0.16 | 31.88±0.76 |
| HL | fencing | HL$_{fencing}$ | 120.16 | 49.44 | 673.95 | 352.00 | -0.10 | 5.5 | 40.3 | 53.0 | 6.29±0.12 | 37.01±1.95 |
| HL | grazing | HL$_{grazing}$ | | | | | | 14.2 | 24.6 | 59.2 | 6.43±0.08 | 29.04±1.81 |
| XL | fencing | XL$_{fencing}$ | 116.74 | 43.60 | 1198.22 | 263.50 | 3.50 | 5.9 | 8.7 | 83.4 | 6.65±0.16 | 12.27±1.03 |
| XL | grazing | XL$_{grazing}$ | | | | | | 8.6 | 5.0 | 84.5 | 6.60±0.19 | 9.00±0.60 |
| XH | fencing | XH$_{fencing}$ | 114.09 | 42.37 | 1224.96 | 270.60 | 4.20 | 4.1 | 5.8 | 75.9 | 7.40±0.15 | 7.84±0.65 |
| XH | grazing | XH$_{grazing}$ | | | | | | 4.8 | 10.9 | 75.0 | 7.65±0.13 | 7.02±0.80 |

## 2 Materials and Methods

### 2.1 Soil sampling

In August 2021, 10 soils were sampled from 5 temperate grasslands of the Inner Mongolian Plateau, China (Table 1). Before sampling, we measured the plant aboveground biomass using the dry weighing method. At each site, soils of the top 20 cm layer were sampled from continuous grazing grassland and grazing excluded (i.e., fenced) grassland, respectively. Before incubation, all soil samples were passed through a 2 mm sieve to remove visible stones, roots, and other plant debris. After homogenization, soil texture, pH, SOC, and MBC content were measured (See methods below; Table 1 and Fig. S2). All soil samples were stored at -20 °C until the incubation experiment started.

### 2.2 Incubation experiment

For each soil, $^{13}$C-labeled glucose addition treatments were performed and four replicates were conducted. Soil samples equivalent to 20 g air-dried soil were added to 250 ml mason jars. All soils were incubated in the dark at 25 °C and a relative humidity of 60% for 102 days. To maintain soil moisture at 60% water holding capacity (WHC), we added distilled water regularly by measuring the weight changes of the jars, which were covered by a sealing film passable for gases but not water molecules. After a 7-day pre-incubation, $^{13}$C-labeled glucose (99 atom% $^{13}$C, Shanghai Engineering Research Center of Stable Isotope) was added at a dose of 0.4 mg C g$^{-1}$ soil. The glucose solution was prepared by dissolving 0.5 g of glucose in 50 ml of water to make a 10 mg ml$^{-1}$ solution. Further, 2 ml of glucose solution was slowly dripped into the soil using a pipette gun to keep the solution as uniformly distributed in the soil as possible. Correspondingly, 2 ml of water was added to the control. On days 1, 3, 6, 12, 19, 34, 47, 78, and 102 of the incubation, each jar was flushed by $CO_2$-free air for 3 minutes. After that, the $CO_2$ emission rate was measured using an infrared gas analyzer (Li-8100A; Li-COR, USA) within 3 minutes from the headspace. Subsequently, we used the soil $CO_2$ emission data for model calibration and validation. After the last gas measurement, soils were destructively harvested and stored at -80°C for the subsequent measurements.

### 2.3 Measurements of DOC, MBC, POC and MAOC

The chloroform-fumigation-extraction method was used to determine DOC and MBC contents (Vance et al., 1987). One subsample of 5 g fresh soil was fumigated by chloroform in the dark for 24h, and a second subsample (5g) was unfumigated as the control. Soil microbes died after 24 hours of chloroform fumigation, and their cells lysed and released microbial biomass C. The soil was extracted with 0.5M $K_2SO_4$ solution subsequently. The dissolved C in the extracting solution was determined by a rapid CS analyzer (Multi N/C 3100, Analytik jena, Germany). The DOC content was calculated according to the organic C content of unfumigated soil. The MBC content was the difference of DOC between fumigated and unfumigated soils multiplying by the proportionality coefficient of 0.45.

The POC and MAOC content were assessed through the particle size fractionation method, which separates SOC into these two pools. Soil samples (10g) were shaken with 30 mL of sodium hexametaphosphate solution (NaHMP, 50 g L$^{-1}$) at 200 rpm. After 18h, samples were washed with deionized water over a 53 µm sieve in a vibratory shaker (AS 200 control, Retch, Germany)(Sokol et al., 2019b). Both fractions were dried at 65 °C, weighed, and fumigated with hydrochloric acid for 8h to remove inorganic C. Organic C content was determined by an elemental analyzer (rapid CS cube, elementar, Germany). The

C from less than 53 µm fraction was considered MAOC, and the >53 um fraction was considered POC. The experimental design was a trade-off between the incubator's space (which determines the jar volume and soil samples in jars) and the total number of jars. With the limited samples, we decided to do the size separation (i.e., POC vs. MAOC) instead of the density fractionation because we anticipated that the size separation might provide more insights into SOC dynamics and is more related to microbial processes according to the literature (Lavallee et al., 2020). Despite that, both light-POC and heavy-POC

were included in the following modeling analysis to broaden the implication of the experiment.

## 2.4 $^{13}$C partitioning

To analyze the MBC-$^{13}$C concentration, an 8-ml extracting solution from each fumigated and unfumigated soil was freeze-dried, and approximately 8 mg of K$_2$SO$_4$-C was analyzed using an Isotope Ratio Mass Spectrometer (Delta V Advantage, ThermoFisher Scientific, America). The atom% of MBC in control and treated soils was determined using a two-pool mixing

model (Fang et al., 2018):

$$at\%_{MBC} = \frac{at\%_{fumigated} \cdot C_{fumigated} - at\%_{unfumigated} \cdot C_{unfumigated}}{C_{fumigated} - C_{unfumigated}} \, , \qquad (1)$$

where $C_{fumigated}$ and $C_{unfumigated}$ are the C mass in fumigated and unfumigated samples, and $at\%_{fumigated}$ and $at\%_{unfumigated}$ are the C isotope abundance (in atom% $^{13}$C) of the fumigated and non-fumigated samples, respectively.

To analyze the content of POC-$^{13}$C and MAOC-$^{13}$C, approximately 2 mg of wet-sieved and oven-dried soil samples were

determined by the Isotope Ratio Mass Spectrometer. Further, the contributions of glucose-derived C to the DOC, MBC, POC, and MAOC pools were estimated following the isotopic mixing model:

$$C_{glucose-derived} = C_{total} \cdot \frac{at\%_{treatment} - at\%_{soil}}{at\%_{glucose} - at\%_{soil}} \, , \qquad (2)$$

$$C_{soil} = C_{total} - C_{glucose-derived} \, , \qquad (3)$$

Where $at\%_{treatment}$, $at\%_{soil}$, $at\%_{glucose}$ are the C isotope compositions (in atom% $^{13}$C) of the glucose-treated soil,

original soil, and added glucose, respectively; $C_{glucose-derived}$, $C_{soil}$ and $C_{total}$ are the glucose-derived, soil-derived C and total SOC content (mg C g$^{-1}$ soil) in the glucose-treated soil, respectively.

## 2.5 Modeling analysis

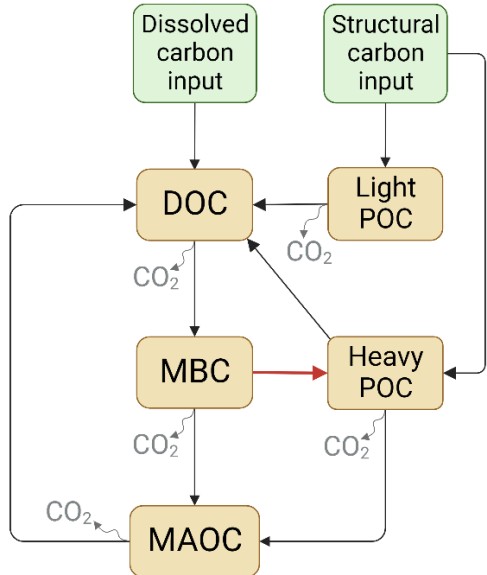

**Figure 1: The model scheme of soil carbon (C) dynamics.** Model I and Model II share similar structure except that Model II includes a C flow from MBC to heavy-POC (red arrow) but Model I does not.

The SOC dynamics were simulated using two mechanistic models (Fig. 1). Models designed based on the data of soil C pools and $CO_2$ emission fluxes in the incubation experiment. Here, it is important to note that in both soil C models, POC is divided into two pools: the light-POC and the heavy-POC. This is because heavy-POC is a plant residue-microbial product-soil mineral complex, which is more likely to be the destination of dissolved C inputs than light-POC, which only comprises plant residues (Samson et al., 2020). Therefore, the two POC pools were modeled separately. In addition, most parts of the two models were identical except that Model I did not include the C flow from MBC to heavy-POC, but Model II did. Model I assumed that plant structural residues were the only POC source, whereas Model II assumed that heavy-POC could be from both plant and microbial residues. Thus, dissolved C can be transformed to heavy-POC via microbial metabolism in Model II. The two models shared a similar structure:

$$\frac{dX(t)}{dt} = AKX(t) \ ,$$ (4)

where

$$X(t) = \begin{bmatrix} x_D \\ x_B \\ x_H \\ x_L \\ x_M \end{bmatrix},$$ (5)

and

$$K = \begin{bmatrix} k_D & & & & \\ & k_B & & & \\ & & k_H & & \\ & & & k_L & \\ & & & & k_M \end{bmatrix},$$ (6)

In Model I,

$$A = \begin{bmatrix} -1 & & f_{DH} & f_{DL} & f_{DM} \\ f_{BD} & -1 & & & \\ & & -1 & & \\ & & & -1 & \\ f_{MB} & f_{MH} & & -1 \end{bmatrix},$$ (7)

whereas in Model II

$$A = \begin{bmatrix} -1 & & f_{DH} & f_{DL} & f_{DM} \\ f_{BD} & -1 & & & \\ & f_{HB} & -1 & & \\ & & & -1 & \\ f_{MB} & f_{MH} & & -1 \end{bmatrix},$$ (8)

In matrix $X$, $x_D$, $x_B$, $x_H$, $x_L$ $x_M$ are the pool sizes of DOC, MBC, heavy-POC, light-POC and MAOC, and $k_D$, $k_B$, $k_H$,

$k_L$, $k_M$ in matrix K are their turnover rates, respectively. In matrix A, $f_{BD}$ means the fraction transfer from the DOC pool to the MBC pool, other transfer coefficients $f$ represent in the same way (See details in Table 2). The measured DOC and MBC before incubation were used as their respective initial pool sizes, whereas a to-be-determined parameter $f_{heavy-POC}$ was used to represent the initial fraction of heavy-POC – i.e., $initial\ POC = (SOC - DOC - MBC) \times f_{heavy-POC}$. Correspondingly, the initial light-POC pool size was calculated as $(SOC - DOC - MBC) \times f_{light-POC}$, and the initial MAOC pool size was

calculated as $(SOC - DOC - MBC) \times (1 - f_{heavy-POC} - f_{light-POC})$. Overall, Model I had 13 and Model II had 14 to-be-determined parameters (Table 2). Because the glucose addition was [13]C-labelled, each C pool was further divided into soil-derived and glucose-derived pools. We considered all glucose addition entered the glucose-derived DOC pool in the beginning.

**Table 2: Description of the soil carbon (C) model parameters.**

| Parameter | Description | Unit |
|---|---|---|
| $f_{heavy-POC}$ | Initial fraction of the heavy-POC pool | - |
| $f_{light-POC}$ | Initial fraction of the light-POC pool | - |
| $k_D$ | Turnover rate of the DOC pool | mg C g$^{-1}$ soil h$^{-1}$ |
| $k_B$ | Turnover rate of the MBC pool | mg C g$^{-1}$ soil h$^{-1}$ |
| $k_H$ | Turnover rate of the heavy-POC pool | mg C g$^{-1}$ soil h$^{-1}$ |
| $k_L$ | Turnover rate of the light-POC pool | mg C g$^{-1}$ soil h$^{-1}$ |
| $k_M$ | Turnover rate of the MAOC pool | mg C g$^{-1}$ soil h$^{-1}$ |
| $f_{BD}$ | DOC to MBC transfer coefficient | - |
| $f_{MB}$ | MBC to MAOC transfer coefficient | - |
| $f_{DM}$ | MAOC to DOC transfer coefficient | - |
| $f_{DL}$ | Light-POC to DOC transfer coefficient | |
| $f_{DH}$ | Heavy-POC to DOC transfer coefficient | |
| $f_{MH}$ | Heavy-POC to MAOC transfer coefficient | - |
| $f_{HB}$ | MBC to heavy-POC transfer coefficient (Only exist in model II) | - |

The models were calibrated using soil C pools and $CO_2$ emission rate data through the adaptive Metropolis algorithm (Haario et al., 2001; Hararuk et al., 2014). For the POC pool, heavy-POC pool and light-POC pool were summed together for the calibration. The $CO_2$ emission data were divided into two groups: 7 out of the 9 flux measurements for each soil were randomly selected for the model calibration, while the other 2 measurements were used for the model validation. The prior probability density functions (PDFs) were assumed as uniform distributions over parameter ranges based on previous studies (Li et al., 2014; Liang et al., 2015). The parameters' posterior PDFs were proportional to the prior PDFs and a cost function from data. The cost function was calculated as:

$$P(Z \mid \theta) \propto \exp\left\{ -\sum_{t \in \text{obs}(Z)} \frac{[O_f(t) - M_f(t)]^2}{2\sigma_f^2(t)} - \sum_{i \in \text{obs}(Z)} \frac{[O_p(i) - M_p(i)]^2}{2\sigma_p^2(i)} \right\} , \qquad (9)$$

where $t$ denotes the measurement time of fluxes and $i$ denotes C pools. $\sigma^2$ is the standard deviation of measurements. $O_f$ and $M_f$ are the observed and modeled $CO_2$ emission fluxes. $O_p$ and $M_p$ are the observed and modeled values of C pools. After the model calibration and validation, we randomly selected 100 sets of parameters from posterior PDFs of the adaptive Metropolis algorithm for further modeling experiments. For each model, we set up two C input scenarios. To fit with the incubation experiment including dissolved C input only, we set up the scenario of "DOC input only." To make the prediction closer to the natural C input in the field, we set up the scenario of "DOC+POC input." And the amount of C input was

approximately equivalent to local annual C influxes (Table S1). The calibrated models were run for 500 years to steady states to compare the modeled SOC change under different scenarios. After that, the models were run along a gradient of C input increase from 1% to 20% with a 1% interval to reach another steady state. Then the impact of C flow from MBC to heavy-POC (i.e., $f_{HB}$) on long-term SOC sequestration was assessed by comparing the behaviors of SOC dynamics between Model I and Model II.

**2.6 Statistical analysis**

The two-way analysis of variance (ANOVA) was used to reveal the effects of sites, fencing, and their interaction on plant aboveground biomass, initial MBC, SOC, soil texture (Table S2), and glucose-derived SOC, MAOC, POC, MBC, DOC, and cumulative respiration (Table S3). The differences between fencing and grazing treatment and the differences caused by $f_{HB}$ between Model I and Model II were tested using the one-way ANOVA. All data were separately tested for normality using the Shapiro–Wilk test and for homoscedasticity using Bartlett's test in advance. In cases where the assumptions of normality or homoscedasticity were not met, a reciprocal transformation was applied to the original data, and analyses were carried out on the transformed data. In cases where the reciprocal transformed data did not meet the test requirements, the Kruskal-Wallis test was applied. The difference was considered statistically significant at the level of $P < 0.05$. The statistical analyses were performed in R 4.1.2. The model was performed in Matlab 2021a.

**3 Results**

**3.1 Effects of fencing and sites on new C formation**

Analysis of different soils and plant investigation data showed that fencing and sites significantly affect plant aboveground biomass, MBC, SOC, and soil texture (Table S2). Generally, plant aboveground biomass, MBC, and SOC were significantly increased after fencing (Fig. S1, S2). For the new C formation, sites had significant effects on the formationof each C pool and respiration, in which glucose-derived MAOC and POC at HL site was significantly higher than that at other sites (Table S3, Fig. S3). Fencing also significantly affected the amount of glucose C entering MAOC as well as the cumulative soil respiration, in which fencing soils show a lower amount of MAOC formationand higher soil respiration (Table S3, Fig. S3, S4).

**3.2 Effects of dissolved carbon inputs on new C formation**

Across the 10 soils, 84.28 –175.80 mg kg$^{-1}$ soil of the glucose C (equivalent to 21.07% – 43.95% of the initial glucose addition) retained in the soil after the 102-day incubation, among which 1.58% – 28.00%, 48.73% – 75.51%, 20.34% – 35.80% of retained glucose C distributed in POC, MAOC, and MBC, respectively (Fig. 2). At the end of incubation, the proportion of glucose-derived C to total POC was 0.16% ~ 0.67% and to total MAOC was 0.26% ~ 1.46%. Additionally, glucose-derived

MAOC and POC were correlated with glucose-derived MBC (Fig. 3a). Furthermore, glucose-derived MAOC and POC increased with the fraction of clay and silt ($R^2 = 0.62$ and 0.92, respectively, Fig. 3b).

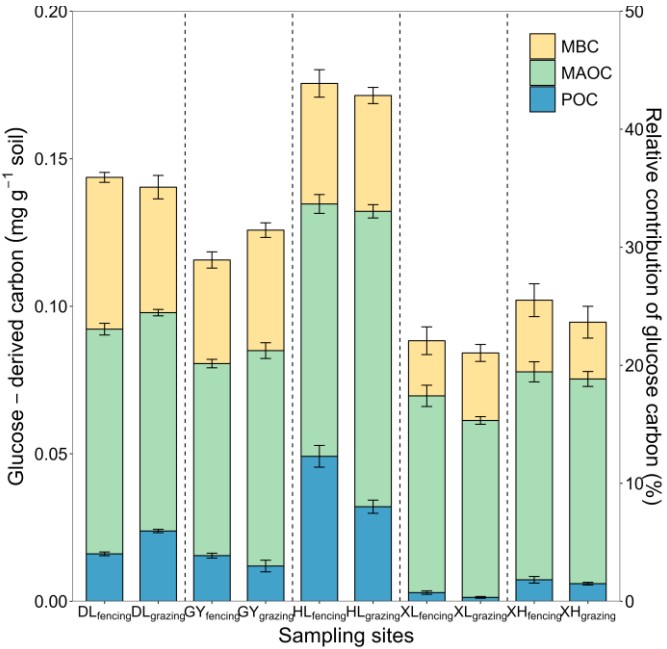

**Figure 2: Distributions of glucose-derived C in soil C pools.** microbial biomass C: MBC, mineral-associated organic C: MAOC, particulate organic C: POC. The left y-axis is absolute amounts of glucose C into MAOC, POC and MBC pools. The right y-axis is relative contribution of newly stabilized C to total glucose C input. The error bars represent the standard errors of four replicates. The vertical dashed line divides the x-axis into five sampling sites, each with fencing treatment in the first column and grazing treatment in the second column.

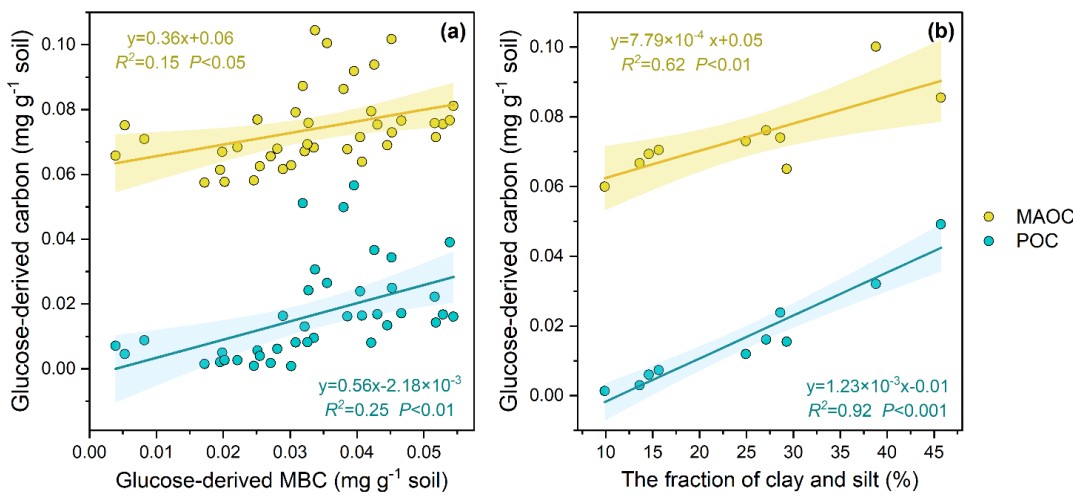

**Figure 3: Correlation of glucose-derived POC and MAOC on MBC (a) and soil texture (b).** Shaded areas represent the 95% confidence intervals for the regression lines.

### 3.3 Modeling analysis of soil C dynamics

Compared with Model II, the absence of the $f_{HB}$ in Model I affects other parameters differently (Table S5). On average,
compared to Model II, Model I showed greater $k_L$, $k_M$, $f_{BD}$, $f_{MB}$, $f_{DM}$, but smaller $k_D$, $k_B$, $k_H$, $f_{DL}$, $f_{DH}$, $f_{MH}$. Although both models fitted respiration flux data well (Fig. S5), Model I, without the dissolved C flow from MBC to POC, was not able to reproduce the observed glucose-derived POC (Fig. S6). At the steady state, when C input only included DOC (dissolved C input only), SOC content in Model I was 10.04% – 53.52% less than that in Model II ($P < 0.05$; Fig. 4). When C input was from both DOC and POC (dissolved and structural C input), excluding dissolved C flow from MBC to POC in Model I decreased SOC content up to 48.02% compared to Model II ($P < 0.05$; Fig. 4). The effect of microbe-derived POC on SOC
sequestration still existed when C input increased. Along with the C input gradient, the SOC difference between the two models was enlarged (Fig. S7). When DOC input increased by 20%, the SOC increases (normalized to their respective steady state) were 0.08 – 4.40 Mg C ha$^{-1}$ soil in Model I and 0.13 – 9.53 Mg C ha$^{-1}$ soil and Model II ($P < 0.05$, Fig. S8). Similarly, when both DOC and POC input increased by 20%, Model II produced a significantly greater SOC content than Model I (0.31 – 18.47 Mg C ha$^{-1}$ soil by Model II vs. 0.21 – 12.55 Mg C ha$^{-1}$ soil by Model I; $P < 0.05$, Fig. S8).

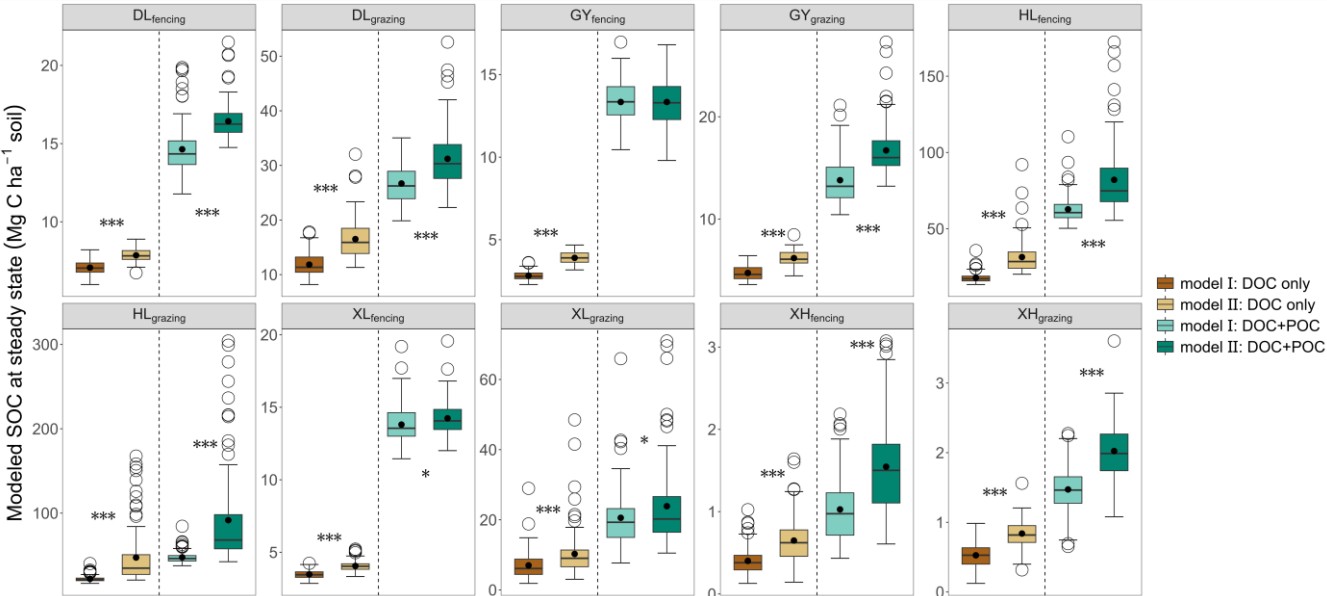


**Figure 4: Modeled SOC content at steady state under two types of C input conditions.** The two different C input scenarios for each site are separated by a dotted line. The upper and lower ends of boxes denote the 0.25 and 0.75 percentiles, respectively.

The solid line and dot in the box mark the median and mean of each dataset. The open circles denote outliers. Asterisks represent significant differences between Model I and Model II (*$P < 0.05$, **$P < 0.01$, ***$P < 0.001$).

## 4 Discussion

### 4.1 Microbe-mediated dissolved C flow to the POC pool

This study showed that most of labile C preferentially entered the MAOC pool, but still up to 12.29% of the glucose C has become part of POC after the 102-day incubation. The results indicate that dissolved labile plant compounds (glucose in our case), in addition to structural litter, could be a significant contributor to POC. Linear regression analyses indicate that glucose C can enter the POC pool via multiple pathways (Craig et al., 2022). Specifically, glucose-derived POC is positively correlated with the glucose-derived MBC (Fig. 3a), suggesting that the transformation of glucose to POC could be dependent on the microbe-mediated biochemical pathway. Meanwhile, glucose-derived POC is positively correlated with the fraction of clay and silt as well ($R^2 = 0.92$, Fig. 3b), further indicating that dissolved C entering into POC is an interaction of physical and biochemical processes. These results are consistent with previous studies, which showed the formation of heavy-POC (or coarse-MAOC) from microbial by-products binding with the silt- and clay-sized soil minerals (Samson et al., 2020). Our results are supported by previous studies with images from scanning electron microscopy (SEM) and nano-scale secondary ion mass spectrometry (NanoSIMS), which show that microorganisms could absorb to the surface of particulate organic matter (POM) and bind it with mineral (Kopittke et al., 2020; Witzgall et al., 2021). Meanwhile, the higher clay and silt content means the more microaggregates and more POC protected from decomposition (Wang et al., 2003). These results were used to support the model structure that we next use for prediction, whereby dissolved C inputs enter the heavy-POC pool under the processes of microbes.

The result that labile C can enter the POC pool is partially inconsistent with the two-pathway framework, which proposes that low-molecular-weight, water-soluble inputs contribute primarily to MAOC formation via the microbe-mediated biochemical pathway, whereas POC is formed primarily from the polymeric structural inputs via the physical transfer pathway (Cotrufo et al., 2015). Our results, combined with previous studies, demonstrate that the biochemical and physical pathways in SOC formation may not be independent with each other. Rather, the formations of MAOC and POC are continuous through close interactions of physical and microbial processes, during which POC originated from dissolved substrates is a critical component in SOC dynamics.

### 4.2 Effect of dissolved substrates-originated POC on SOC sequestration

As POM surfaces are considered the hotspots of microbial activities and the cores of aggregate formation (Tisdall and Oades, 1982; Witzgall et al., 2021), our modeling analyses indicated that dissolved substrates-originated POC can significantly influence long-term SOC sequestration. Although both Model I and Model II fitted the C flux data well, Model I, which does

not include the dissolved C flow from MBC to POC, was not able to reproduce the observed POC changes (Fig. S6). The results emphasize the potential usefulness of including the process of dissolved C flow to POC in SOC dynamic models. During the model calibration, including or not the dissolved C flow from MBC to POC significantly affected the estimations of turnover and transfer parameters. Specifically, the absence of the $f_{HB}$ enabled more C flow into the MAOC pool, the algorithm tended to mistakenly elevate the turnover rate of MAOC by 12.28% in order to fit the C pool data in short-term incubation. While this does not have a great impact on the short-term data fitting process, it can significantly affect the long-term SOC predictions. As a result, the absence of the mechanism of microbe-mediated dissolved C flow to POC leads to an underestimation of SOC sequestration in Model I (Fig. 4). In addition, the underestimation of SOC sequestration would be proportionally exacerbated as the magnitude of C input increases (Fig. S7, S8). These results indicate that the process of microbe-mediated dissolved C flow to POC is critical for long-term SOC sequestration and should be considered in soil C dynamic models.

## 4.3 Fencing effect on new C formation and soil respiration from incubation experiment

An additional goal of our study was to explore the mechanisms of soil C formation after the fencing management in grassland ecosystems. Many research suggests that appropriate grazing exclusion by fencing in degraded grassland can increase soil C storage, promoting restoration (Bardgett et al., 2021; Lu et al., 2018). Our field results showed that fencing sites had greater SOC and MBC contents (Table S2, Fig. S2). This can be attributed to the increased C input, which stimulates microbial growth and allows more C to stabilize in the SOC pool (Table S2, Fig. S1). However, in the incubation experiment, fencing soils showed greater cumulative respiration and lower MAOC formation (Fig. S3 and S4). These inconsistent results between the field observations and the incubation experiment suggest that the increased SOC sequestration by fencing could be primarily due to the C input instead of the C transformation in the soil. Specifically, the observed increases in soil C stocks of fenced grasslands were closely related to the increased plant production and C inputs from grazing exclusion (Fig. S1). Once the C input kept consistent between fencing and grazing soils, multiple linear regression showed that the predictor variable of clay and silt content explained 91.85% of the variance in new SOC formation (Table S4). Additionally, the clay and silt content also dominated the magnitude of soil C formation across sites (Fig. 3b). Meanwhile, higher cumulative respiration in fencing soils can be explained by initial SOC and soil texture, presenting a positive effect of higher SOC content but negative effect of clay and silt content (Table S4). Moreover, no significant difference of glucose-derived MBC was observed between fencing and grazing soils (Fig. S3c), which further validates that C input is the dominant factor influencing soil microorganisms.

## 5 Conclusions

This study provides direct evidence that dissolved C input can not only enter MAOC, but also POC through the microbe-mediated biochemical pathway. As a result, dissolved plant compounds, in addition to structural litter, are vital contributors to POC. The microbe-mediated dissolved C flow to POC is a critical component in SOC dynamics. From the modeling

perspective, ignoring the mechanism of microbe-mediated dissolved C flow to POC would cause a significant underestimation
295  of long-term SOC sequestration.

## Author contributions

Junyi Liang and Qintana Si designed the study. Yaowen Zhang, Xun Sun conducted the soil sampling. Qintana Si, Kangli Chen, and Bin Wei conducted the incubation experiment. Junyi Liang and Qintana Si developed the modeling framework. Junyi Liang and Qintana Si performed the analyses. All the authors contributed to writing the manuscript.

## Competing interests

300

The contact author has declared that none of the authors has any competing interests.

## Acknowledgements

This study was financially supported by the National Natural Science Foundation of China (42203077, 32192462), the Chinese Universities Scientific Fund (2020RC009) and the 2115 Talent Development Program of China Agricultural University (1201-
305  336 00109017). The authors declare that they have no conflict of interest.

## Data availability statement

All data are freely available at **https://doi.org/10.6084/m9.figshare.24773205.v1**.

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
