# Peer review of "Dissolved carbon flow to particulate organic carbon enhances soil carbon sequestration"

_EGUsphere, 2023_

## Author Comment (AC1)

**Revised Table S2.** Results of the two-way analysis of variance (ANOVA) on the effects of sites, fencing, and their interaction (sites: fencing) on vegetation aboveground biomass, initial MBC, SOC, and soil mineral.

| | Df | Sum Sq | $F$-value | $P$-value |
|---|---|---|---|---|
| **Aboveground biomass** | | | | |
| Sites | 4 | $1.52 \times 10^5$ | $2.14 \times 10^1$ | <0.001 |
| Fencing | 1 | $2.66 \times 10^5$ | $1.50 \times 10^2$ | <0.001 |
| Sites: Fencing | 4 | $1.20 \times 10^5$ | $1.69 \times 10^1$ | <0.001 |
| Residuals | 50 | $8.87 \times 10^4$ | | |
| **Initial MBC** | | | | |
| Sites | 4 | 1.40 | $7.17 \times 10^1$ | <0.001 |
| Fencing | 1 | $2.92 \times 10^{-2}$ | 5.98 | <0.05 |
| Sites: Fencing | 4 | $3.04 \times 10^{-2}$ | 1.56 | 0.211 |
| Residuals | 30 | $1.46 \times 10^{-1}$ | | |
| **Initial SOC** | | | | |
| Sites | 4 | $1.04 \times 10^4$ | $5.36 \times 10^2$ | <0.001 |
| Fencing | 1 | $9.40 \times 10^1$ | $1.94 \times 10^1$ | <0.001 |
| Sites: Fencing | 4 | $1.31 \times 10^2$ | 6.74 | <0.01 |
| Residuals | 20 | $9.70 \times 10^1$ | | |
| **Soil mineral (Silt and clay)** | | | | |
| Sites | 4 | $3.67 \times 10^3$ | $3.70 \times 10^2$ | <0.001 |
| Fencing | 1 | $1.60 \times 10^1$ | 6.32 | <0.05 |
| Sites: Fencing | 4 | $4.80 \times 10^1$ | 4.87 | <0.05 |
| Residuals | 10 | $2.50 \times 10^1$ | | |

**Revised Table S3.** Results of the two-way ANOVA on the effects of sites, fencing, and their interaction (sites: fencing) on glucose-derived SOC, MAOC, POC, MBC, DOC, and cumulative respiration. Cumulative respiration is the total respiration after 102d incubation calculated from the respiration rate. The SOC content is the sum of the sizes of the other four C pools combined.

| | Df | Sum Sq | $F$-value | $P$-value |
|---|---|---|---|---|
| **Glucose-derived SOC** | | | | |
| Sites | 4 | $3.85\times10^{-2}$ | $1.10\times10^{2}$ | **<0.001** |
| Fencing | 1 | $9.00\times10^{-5}$ | $9.75\times10^{-1}$ | 0.331 |
| Sites: Fencing | 4 | $9.00\times10^{-4}$ | 2.58 | 0.058 |
| Residuals | 30 | $2.62\times10^{-3}$ | | |
| **Glucose-derived MAOC** | | | | |
| Sites | 4 | $3.94\times10^{-3}$ | $3.95\times10^{1}$ | **<0.001** |
| Fencing | 1 | $3.18\times10^{-4}$ | $1.27\times10^{1}$ | **<0.01** |
| Sites: Fencing | 4 | $4.78\times10^{-4}$ | 4.79 | **<0.01** |
| Residuals | 30 | $7.48\times10^{-4}$ | | |
| **Glucose-derived POC** | | | | |
| Sites | 4 | $7.15\times10^{-3}$ | $1.74\times10^{2}$ | **<0.001** |
| Fencing | 1 | $4.00\times10^{-5}$ | 3.85 | 0.059 |
| Sites: Fencing | 4 | $7.78\times10^{-4}$ | $1.90\times10^{1}$ | **<0.001** |
| Residuals | 30 | $3.08\times10^{-4}$ | | |
| **Glucose-derived MBC** | | | | |
| Sites | 4 | $4.43\times10^{-3}$ | $1.83\times10^{1}$ | **<0.001** |
| Fencing | 1 | $6.00\times10^{-6}$ | $1.02\times10^{-1}$ | 0.752 |
| Sites: Fencing | 4 | $2.28\times10^{-4}$ | $9.37\times10^{-1}$ | 0.456 |
| Residuals | 30 | $1.82\times10^{-3}$ | | |
| **Glucose-derived DOC** | | | | |
| Sites | 4 | $1.62\times10^{-7}$ | 8.89 | **<0.001** |
| Fencing | 1 | $3.42\times10^{-8}$ | 7.50 | **<0.05** |
| Sites: Fencing | 4 | $5.08\times10^{-8}$ | 2.78 | **<0.05** |
| Residuals | 30 | $1.37\times10^{-7}$ | | |
| **Cumulative respiration** | | | | |
| Sites | 4 | $1.21\times10^{3}$ | $2.05\times10^{2}$ | **<0.001** |
| Fencing | 1 | 8.40 | 5.67 | **<0.05** |
| Sites: Fencing | 4 | $1.38\times10^{1}$ | 2.33 | 0.079 |
| Residuals | 30 | $4.44\times10^{1}$ | | |

**Revised Table S4.** Multiple linear regression of cumulative respiration and glucose-derived SOC. Cumulative respiration is the total respiration after 102d incubation calculated from the respiration rate. Soil mineral was represented by the sum of clay and silt content.

| | Estimate | Std.Error | t value | P |
|---|---|---|---|---|
| Cumulative respiration~soil mineral + SOC, Multiple $R^2$=0.8743, Adjusted $R^2$=0.8384 | | | | |
| Intercept | 6.229 | 1.842 | 3.382 | <0.05 |
| soil mineral | -0.381 | 0.088 | -4.349 | <0.01 |
| SOC | 0.357 | 0.051 | 6.978 | <0.001 |
| Glucose-derived SOC~soil mineral, Multiple $R^2$=0.9185, Adjusted $R^2$=0.9083 | | | | |
| Intercept | 0.056 | 0.008 | 7.203 | <0.001 |
| soil mineral | 0.003 | 0.000 | 9.497 | <0.001 |

[Figure]

**Revised Figure 1. The model scheme of soil carbon (C) dynamics.** Model I and Model II share similar structure except that Model II includes a C flow from MBC to heavy-POC (red arrow) but Model I does not.

[Figure]

**Revised Figure 4: Modeled SOC content at steady state under two types of C input conditions.** The upper and lower ends of boxes denote the 0.25 and 0.75 percentiles, respectively. The solid line and solid dots in the box mark the median and mean of each dataset. Hollow dot denotes outliers. Asterisks represent significant differences between Model I and Model II (*P < 0.05, **P < 0.01, ***P < 0.001).

[Figure]

**Revised Figure S1. Response of above ground biomass to sites and fencing.** Error bars represent the standard errors.

[Figure]

**Revised Figure S2. Response of initial SOC and MBC to sites and fencing.** Error bars represent the standard errors in bar graph. In the boxplot, the upper and lower ends of boxes denote the 0.25 and 0.75 percentiles, respectively. The solid line and solid dots in the box mark the median and mean of each dataset. Hollow dot denotes outliers. Asterisks represent significant differences between grazing and fencing (*$P < 0.05$, **$P < 0.01$).

[Figure]

**Revised Figure S3**. **Response of C sequestration to sites and fencing. a**. Glucose-derived MAOC; **b**. Glucose-derived POC; **c**. Glucose-derived MBC; **d**. Glucose-derived DOC. The data are the means of four replicates and the error bars represent the standard errors of four replicates. Different letters above bars represent significant differences among sites ($P <$ 0.05). Asterisks represent significant differences between grazing and fencing treatment (*$P <$ 0.05, **$P <$ 0.01, ***$P <$ 0.001).

[Figure]

**Revised Figure S4. Response of soil cumulative respiration to sites and fencing.** The data
are the means of four replicates and the error bars represent the standard errors of four
replicates. The first two capital letters in the legend are the initials of the site, followed by
"in" for grazing excluded grassland and "out" for grazing grassland.

[Figure]

**Revised Figure S5. Comparison of observed and modeled CO₂ flux in the incubation experiment.** Diagonal black dotted line is 1:1 line.

[Figure]

**Revised Figure S6. Observed and modeled soil C pools by Model I and II.** The error bars represent the standard deviation of each dataset.

[Figure]

**Figure R1. MAOC and POC contents of glucose-added and control treatments at the end of incubation.** The top half of the figure represents MAOC, and the bottom half represents POC. filled bars represent the glucose-added treatment and open bars represent the control. Mean ± SD.

---

## Author Response (AR1)

**Letter of Responses**

Author Note: Comments from editors and reviewers are in black and the responses follow in blue. Line numbers in the responses are those in the marked-up revision. Reviewers' comments have been numbered for the easy reading.

RC1: 'Comment on egusphere-2023-1483', Anonymous Referee #1, 25 Aug 2023

**General comments:**

Comment 1: The preprint under review explores the dynamics of microbial carbon contributions to particulate organic carbon (POC) in grassland ecosystems. The study investigates the relationship between microbial-derived carbon from labile sources, like glucose, and its incorporation into POC. While the manuscript offers valuable insights into the intricate connections between microbial processes and POC formation, there are several aspects that warrant attention and further clarification.

Response: We greatly appreciate the constructive comments, which are very helpful to improve the manuscript. Point-by-point responses to comments are detailed below.

Comment 2: In the introduction, the authors could enhance the clarity of their hypotheses by providing more concise predictions based on the research questions. This would provide readers with a clearer roadmap of what to expect in the subsequent sections. Additionally, the intro raises a pertinent point regarding the heavy POC or coarse MAOC fractions, but this fractionation scheme was not utilized in the study. The use of density fractionation to isolate this pool could potentially yield more targeted insights.

Response: We agree with the reviewer that including hypotheses would enhance the clarity. We have added our hypotheses in the revision (line 79-82): "*To answer the questions, we had three hypotheses. First, dissolved C can get into the POC pool in addition to the MAOC pool due to interactions between soil physical and biochemical processes. Second, the rate of POC conversion from glucose is dependent upon microbial activity due to the land use change across sites. Finally, adding the pathway from dissolved C input to the POC pool can promote microbial C use efficiency, further enhancing SOC sequestration.*"

We also appreciate the reviewer's suggestion on including the heavy POC or coarse MAOC fractions in the analysis. In the revision, we have modified the model structures in the modeling experiment. In the revised model, the POC pool has been divided into heavy-

POC and light-POC (revised Fig.1).

Correspondingly, we have revised the Materials and Methods section to describe the new model (line 135-139): "*The SOC dynamics were simulated using two mechanistic models. Most parts of the two models were identical except that Model I did not include the C flow from MBC to heavy-POC, but Model II did (Fig. 1). Model I assumed that plant structural residues were the only POC source, whereas Model II assumed that heavy-POC could be from both plant and microbial residues. Thus, dissolved C can be transformed to heavy-POC via microbial metabolism in Model II.*"

The new model structure simulates the experiential data well for the 10 sites (Fig. S5 and S6). With the well-trained model, we have conducted the modeling experiment, which has shown similar scenario predictions with the original models (revised Fig. 4, S7 and S8). The results confirms that glucose-derived carbon input enters the heavy-POC pool is convincing.

The reviewer might wonder why we did not do the density fractionation experimentally. The reason was we were not able to do both density and size fractionations due to the limited samples. In the experiment, the soil sample incubated for each replicate was equivalent to 20 g air-dried soil. The experimental design was a trade-off between the incubator's space (which determines the jar volume and soil samples in jars) and the total number of jars. With the limited samples, we measured microbial biomass C (MBC), dissolved organic C (DOC), POC, MAOC, as well as their $^{13}C/^{12}C$ ratio. We decided to do the size separation (i.e., POC vs. MAOC) instead of the density fractionation because we were anticipating that the size separation may provide more insights into SOC dynamics and is more related with microbial processes according to the literature (Lavallee et al., 2020).

Although we did not have enough samples to do both density and size fractionations experimentally, we do agree with the reviewer that including the heavy POC or coarse MAOC fractions in the analysis would potentially yield more targeted insights. To that end, we have made thorough revision to the modeling analysis. We hope the revised modeling analysis has addressed the reviewer's concern regarding the heavy POC or coarse MAOC fractions.

[Figure]

**Revised Figure 1: The model scheme of soil carbon (C) dynamics.** Model I and Model II share similar structure except that Model II includes a C flow from MBC to heavy-POC (red arrow) but Model I does not.

[Figure]

**Revised Figure 4: Modeled SOC content at steady state under two types of C input conditions.** The two different C input scenarios for each site are separated by a dotted line. The upper and lower ends of boxes denote the 0.25 and 0.75 percentiles, respectively. The solid line and dot in the box mark the median and mean of each dataset. The open circles denote outliers. Asterisks represent significant differences between Model I and Model II (*$P < 0.05$, **$P < 0.01$, ***$P < 0.001$).

[Figure]

**Revised Figure S7. Continuous relative changes in the steady state along a gradient of C input increase under two types of C input conditions.** C input increase from 1% to 20% with a 1% interval.

[Figure]

**Revised Figure S8. Relative changes in steady state at 20% C input increase under two types of C input conditions.** The upper and lower ends of boxes denote the 0.25 and 0.75 percentiles, respectively. The solid line and solid dots in the box mark the median and mean of each dataset. Hollow dot denotes outliers. Asterisks represent significant differences between Model I and Model II (*$P < 0.05$, **$P < 0.01$, ***$P < 0.001$).

Comment 3: The collection of grazed versus ungrazed soil samples is mentioned in the methods section, yet its rationale remains absent from both the introduction and results. Addressing the purpose of this comparison within the context of the research objectives would be beneficial for readers' comprehension.

Response: The grazed vs. ungrazed (i.e., fenced) soil samples were collected to test whether the land use change has any effect on the destinations of glucose-derived C. The results showed that the land use change had inconsistent effects, which were weakened in the previous submission to avoid possible distraction of the key points. But we do agree with the reviewer that addressing the purpose of this comparison within the context of the research objectives would be beneficial for readers' comprehension. In the revision, we have added new sections to Introduction (lines 65-71), Results (lines 192-198), and Discussion (lines 277-292), to further discuss more about fencing effect on C sequestration. We also added figures of vegetation aboveground biomass (revised Fig. S1), initial SOC and MBC (revised Fig. S2), and soil respiration (revised Fig. S4) to provide comprehensive information of our study to the readers.

Line 65-71 in Introduction: "*Meanwhile, the soil C dynamics are sensitive to land use changes (Del Galdo et al., 2003; Grandy and Robertson, 2007). Overgrazing and conversion of grasslands to farmlands have resulted in significant ecosystem degradation in the grasslands of northern China (Wang et al., 2023; Buisson et al., 2022). Fencing is a widely used strategy in order to retard and reverse the grassland degradation. To date, it has been well-studied that fencing can improve the plant community structure of degraded grasslands, increase species diversity, improve soil structure, promote soil microbial biomass and enzyme activity (Lu et al., 2018; Bardgett et al., 2021). However, how differently dissolved substrates affect POC and MAOC dynamics in fencing and grazing grasslands is still unclear.*"

Line 192-198 in Result: "*Analysis of different soils and plant investigation data showed that fencing and sites significantly affect plant aboveground biomass, MBC, SOC, and soil texture (Table S2). Generally, plant aboveground biomass, MBC, and SOC were significantly increased after fencing (Fig. S1, S2). For the new C sequestration, sites had significant effects on the sequestration of each C pool and respiration, in which glucose-derived MAOC and POC at HL site was significantly higher than that at other sites (Table S3, Fig. S3). Fencing also significantly affected the amount of glucose C entering MAOC as well as the cumulative soil respiration, in which fencing soils show a lower amount of MAOC sequestration and higher soil respiration (Table S3, Fig. S3, S4).*"

Line 277-292 in Discussion: "*An additional goal of our study was to explore the mechanisms of soil C sequestration after the fencing management in grassland ecosystems. Many research suggests that appropriate grazing exclusion by fencing in degraded grassland can increase soil C storage, promoting restoration (Bardgett et al., 2021; Lu et al., 2018). Our field results showed that fencing sites had greater SOC and MBC contents (Table S2, Fig. S2). This can be attributed to the increased C input, which stimulates microbial growth and allows more C to stabilize in the SOC pool (Table S2, Fig. S1). However, in the incubation experiment, fencing soils showed greater cumulative respiration and lower MAOC sequestration (Fig. S3 and S4). These inconsistent results between the field observations and the incubation experiment suggest that the increased SOC sequestration by fencing could be primarily due to the C input instead of the C transformation in the soil. Specifically, the observed increases in soil C stocks of fencing grasslands were closely related to the increased plant production and C inputs from grazing exclusion (Fig. S1). Once the C input kept consistent between fencing and grazing soils, multiple linear regression showed that the predictor variable of clay and silt content explained 91.85% of the variance in new SOC sequestration (Table S4). Additionally, the clay and silt content also dominated the magnitude of soil C sequestration across sites (Fig. 3b). Meanwhile, higher cumulative respiration in fencing soils can be explained by initial SOC and soil texture, presenting a positive effect of higher SOC content but negative effect of clay and silt content (Table S4). Moreover, no significant difference of glucose-derived MBC was observed between fencing and grazing soils (Fig. S3c), which further validates that C input is the dominant factor influencing soil microorganisms.*"

[Figure]

**Revised Figure S1**. **Plant aboveground biomass in fencing and grazing grasslands across the five study sites.** Error bars represent the standard errors of six replicates.

[Figure]

**Revised Figure S2**. **Initial SOC (a) and MBC (b) in fencing and grazing grasslands across the five study sites.** Error bars represent the standard errors of three replicates for initial SOC and four replicates for initial MBC in bar graph. In the imbedded boxplot, the upper and lower ends of boxes denote the 0.25 and 0.75 percentiles, respectively. The solid line and dot in the box mark the median and mean of each dataset. The open circles denote outliers. Asterisks represent significant differences between grazing and fencing (*$P < 0.05$, **$P < 0.01$).

[Figure]

**Revised Figure S4. Response of cumulative respiration to sites and fencing.** The error bars represent the standard errors of four replicates. Please see Table 1 for the abbreviations.

Comment 4: Moving to the results section, the reviewer highlights the intriguing findings regarding glucose-derived POC and its proportion within total POC. However, the text does not mention the relative contribution of glucose-derived C to POC and MAOC, which could provide a more holistic perspective.

Response: In the revision, we have refined the relative contribution of glucose-derived C to POC and MAOC in line 200-203: "*Across the 10 soils, 84.28 –175.80 mg kg$^{-1}$ soil of the glucose C (equivalent to 21.07% – 43.95% of the initial glucose addition) retained in the soil after the 102-day incubation, among which 1.58% – 28.00%, 48.73% – 75.51%, 20.34% – 35.80% of retained glucose C distributed in POC, MAOC and MBC, respectively.*"

Comment 5: Furthermore, clarifying whether respiration data solely validated model outputs or served additional purposes would mitigate this ambiguity.

Response: In the original manuscript, we only used the soil respiration data for model

calibration. In the revision, we have added new figures about respiration in revised Fig. S4, and microbial respiration data was not only used as model calibration, but also for the discussion of fencing effects:

Line 194-198: "*For the new C sequestration, sites had significant effects on the sequestration of each C pool and respiration, in which glucose-derived MAOC and POC at HL site was significantly higher than that at other sites (Table S3, Fig. S3). Fencing also significantly affected the amount of glucose C entering MAOC as well as the cumulative soil respiration, in which fencing soils show a lower amount of MAOC sequestration and higher soil respiration (Table S3, Fig. S3, S4).*"

Line 288-292: "*Meanwhile, higher cumulative respiration in fencing soils can be explained by initial SOC and soil texture, presenting a positive effect of higher SOC content but negative effect of clay and silt content (Table S4). Moreover, no significant difference of glucose-derived MBC was observed between fencing and grazing soils (Fig. S3c), which further validates that C input is the dominant factor influencing soil microorganisms.*"

Comment 6: Figure 2 requires clearer labeling, including definitions of "in" and "out" to facilitate interpretation.

Response: For clarity, we have deleted the expressions "in" and "out" and replaced them with "fencing" and "grazing" in all the figures of the revision. And we added more description in the legend of Fig. 2: "*The vertical dashed line divides the x-axis into five sampling sites, each with fencing treatment in the first column and grazing treatment in the second column.*"

Comment 7: The reviewer could identify significant variations in glucose-derived POC and MAOC among different grazing scenarios, which could enrich the discussion by exploring potential explanations for these disparities.

Response: In the revision, we have added new section in Discussion to discuss significant variations in glucose-derived POC and MAOC among different grazing/fencing scenarios (line 281-292): "*However, in the incubation experiment, fencing soils showed greater cumulative respiration and lower MAOC sequestration (Fig. S3 and S4). These inconsistent results between the field observations and the incubation experiment suggest that the increased SOC sequestration by fencing could be primarily due to the C input instead of the C transformation in the soil. Specifically, the*

*observed increases in soil C stocks of fencing grasslands were closely related to the increased plant production and C inputs from grazing exclusion (Fig. S1). Once the C input kept consistent between fencing and grazing soils, multiple linear regression showed that the predictor variable of clay and silt content explained 91.85% of the variance in new SOC sequestration (Table S4). Additionally, the clay and silt content also dominated the magnitude of soil C sequestration across sites (Fig. 3b). Meanwhile, higher cumulative respiration in fencing soils can be explained by initial SOC and soil texture, presenting a positive effect of higher SOC content but negative effect of clay and silt content (Table S4). Moreover, no significant difference of glucose-derived MBC was observed between fencing and grazing soils (Fig. S3c), which further validates that C input is the dominant factor influencing soil microorganisms.*"

Comment 8: The intriguing graphical representation in Figure 3, while interesting, prompts the reviewer to suggest an exploration of heavy POC independently, given its importance in understanding silt/clay and dissolved organic C (DOC) interactions.

Response: We appreciate the constructive comment. Please see our response to Comment 2.

Comment 9: Scaling the C sequestration projections to an ecosystem level, both with and without microbial POC mechanisms, could enhance the practical implications of the findings.

Response: Thanks for the detailed comment. In the revision, we have scaled the projections using the measured soil bulk density from the 10 grasslands. After the scaling, the projected SOC pools at the equilibrium are shown in Mg C ha$^{-1}$ soil (revised Fig. 4).

Comment 10: In the discussion section, addressing the potential direct interaction of glucose-derived carbon with POC, rather than solely through microbial pathways, adds depth to the interpretation. The reviewer aptly acknowledges the significance of the study's findings in challenging the dichotomy between physical and microbial pathways in POC formation. In conclusion, this preprint contributes noteworthy insights into the complex interplay between microbial processes and POC dynamics in grassland ecosystems. Addressing the identified gaps and refining the manuscript in line with the reviewer's suggestions would undoubtedly elevate its scientific impact.

Response: We appreciate the reviewer's recognition of our work and constructive comments. We have revised the manuscript thoroughly. We hope you will find our revision

satisfactory.

**Specific comments:**

Introduction

Comment 11:     Would be nice to see concise hypotheses/predictions for your questions.

Response: We agree that adding assumptions is beneficial and facilitates the integrity of the article. As responded above, we have added our hypotheses in the revision (line 79-82): "*To answer the questions, we had three hypotheses. First, dissolved C can get into the POC pool in addition to the MAOC pool due to interactions between soil physical and biochemical processes. Second, the rate of POC conversion from glucose is dependent upon microbial activity due to the land use change across sites. Finally, adding the pathway from dissolved C input to the POC pool can promote microbial C use efficiency, further enhancing SOC sequestration.*"

Methods

Comment 12:     If you are interested in the heavy POC or coarse MAOC fraction as mentioned in the introduction, why not do a density fractionation to isolate this pool?

Response: We appreciate the insightful comment. As responded to Comment 2 above, we did not have enough samples to conduct both size and density fractionations. But we do agree with the reviewer that including the heavy POC or coarse MAOC fractions in the analysis would potentially yield more targeted insights. To that end, we have modified the model structures in the modeling experiment (revised Fig. 1 and corresponding text in lines 135-139): "*Most parts of the two models were identical except that Model I did not include the C flow from MBC to heavy-POC, but Model II did (Fig. 1). Model I assumed that plant structural residues were the only POC source, whereas Model II assumed that heavy-POC could be from both plant and microbial residues. Thus, dissolved C can be transformed to heavy-POC via microbial metabolism in Model II.*"

Comment 13:     Is there a reason you collected grazed vs ungrazed soils? You do not mention anything about it in your introduction or results.

Response: We appreciate the constructive comment. We have added more description in Introduction, Results, and Discussion to explain why we collected grazing and fencing

soils. Please see our detailed response in Comment 3 above.

Results

Comment 14:     These are very interesting results! It would be helpful to see the data on glucose C remaining as percent C remaining of initial C added. Also, the proportion of total POC that is from glucose C would be good to know. What is the relative contribution of glucose C to POC and MAOC?

Response: In the revision, we added the relative contribution of glucose-derived C to POC and MAOC in revised Fig. 2 in its right y-axis, and added the description in line 200-203: "*Across the 10 soils, 84.28 –175.80 mg kg$^{-1}$ soil of the glucose C (equivalent to 21.07% – 43.95% of the initial glucose addition) retained in the soil after the 102-day incubation, among which 1.58% – 28.00%, 48.73% – 75.51%, 20.34% – 35.80% of retained glucose C distributed in POC, MAOC and MBC, respectively. At the end of incubation, the proportion of total POC that is from glucose C was 0.16% – 0.67%.*"

[Figure]

**Revised Figure 2: Distributions of glucose-derived C in soil C pools.** microbial biomass C: MBC, mineral-associated organic C: MAOC, particulate organic C: POC. The left y-axis is absolute amounts of glucose C into MAOC, POC and MBC pools. The right y-axis is relative contribution of newly stabilized C to total glucose C input. The error bars represent the standard errors of four replicates. The vertical dashed line divides the x-axis into five sampling sites, each with fencing treatment in the first column and grazing treatment in the second column.

Comment 15:  Also, there is no mention of the comparison between sites. For example

Response: We have added a new section in Results to describe the effect of fencing and site specifically (line 192-198): "*Analysis of different soils and plant investigation data showed that fencing and sites significantly affect plant aboveground biomass, MBC, SOC, and soil texture (Table S2). Generally, plant aboveground biomass, MBC, and SOC were significantly increased after fencing (Fig. S1, S2). For the new C sequestration, sites had significant effects on the sequestration of each C pool and respiration, in which glucose-derived MAOC and POC at HL site was significantly higher than that at other sites (Table*

*S3, Fig. S3). Fencing also significantly affected the amount of glucose C entering MAOC as well as the cumulative soil respiration, in which fencing soils show a lower amount of MAOC sequestration and higher soil respiration (Table S3, Fig. S3, S4).*"

Meanwhile, our results showed that soil carbon sequestration varies between sites depending on the heterogeneity of clay and silt content (revised Table 4). As a response, we added more information in the revision (lines 287-288): "*Additionally, the clay and silt content also dominated the magnitude of soil C sequestration across sites (Fig. 3b).*"

**Revised Table S4. Multiple linear regression of cumulative respiration and glucose-derived SOC.** Cumulative respiration is the total respiration for 102d incubation calculated from the respiration rate. Soil texture was represented by the sum of clay and silt content.

|  | Estimate | Std.Error | t value | P |
|---|---|---|---|---|
| Cumulative respiration ~ soil texture + SOC, Multiple $R^2$=0.8743, Adjusted $R^2$=0.8384 |  |  |  |  |
| Intercept | 6.229 | 1.842 | 3.382 | <0.05 |
| soil texture | -0.381 | 0.088 | -4.349 | <0.01 |
| SOC | 0.357 | 0.051 | 6.978 | <0.001 |
| Glucose-derived SOC ~ soil texture, Multiple $R^2$=0.9185, Adjusted $R^2$=0.9083 |  |  |  |  |
| Intercept | 0.056 | 0.008 | 7.203 | <0.001 |
| soil texture | 0.003 | 0.000 | 9.497 | <0.001 |

Comment 16:     You measured respiration, but I do not see any results presented on respiration or glucose derived respiration. Was respiration data only used to validate the model ouputs? This needs more explanation.

Response: Revised as suggested. As responded to Comment 5, we have added new figure (revised Fig. S4) to show respiration data in the revision. and microbial respiration data was not used as model calibration only, but also for the discussion of fencing effects.

Comment 17:     Figure 2. Need description of in and out.

Response: We have modified Fig. 2 and the legends in response to Comment 6. Please find the revised Fig. 2 in the main text and above.

Comment 18:     There are noticeable differences in Glucose derived POC in DLin and DLout,

and differences in glucose derived POC and MAOC between HLin and HLout. These seem like interesting results but are not discussed in Results or Discussion. This seems to be a missed opportunity to discuss why the biogeochemistry might be different under grazing vs no grazing.

Response: We appreciate the detailed comments. According to the one-way ANOVA at each site, there were noticeable differences in glucose-derived POC in $DL_{fencing}$ and $DL_{grazing}$, and differences in glucose-derived POC and MAOC between $HL_{fencing}$ and $HL_{grazing}$ (revised Fig. S3). The two-way ANOVA shows that fencing significantly influenced MAOC sequestration but not POC, and multiple linear regression showed that the process of new C sequestration largely relied on soil texture (revised Table S4). In the revision, we have added more discussion about fencing effect on C sequestration. Please find the detailed response to Comment 3 above.

[Figure]

**Revised Figure S3**. **Response of C sequestration to sites and fencing. a**. Glucose-derived MAOC; **b**. Glucose-derived POC; **c**. Glucose-derived MBC; **d**. Glucose-derived DOC. The data are the means of four replicates and the error bars represent the standard errors of four replicates. Different letters above bars represent significant differences among sites ($P < 0.05$). Asterisks represent significant differences between grazing and fencing treatment (*$P < 0.05$, **$P < 0.01$, ***$P < 0.001$).

Comment 19:     Figure 3. Interesting way to display the data, I like it. These are interesting data. Again, I think this is a missed opportunity to look at heavy POC independently as this is a key fraction in understanding the mechanisms relating silt/clay and DOC.

Response: We agree with the reviewer that heavy-POC is a key fraction in understanding the mechanisms relating silt/clay and DOC. As we responded to Comment 2 and Comment 12 above, we did not have enough samples to conduct both size and density fractionations. In the revision, we have modified the model structures in the modeling experiment (revised Fig. 1 and corresponding text in lines 135-138): "*Most parts of the two models were identical except that Model I did not include the C flow from MBC to heavy-POC, but*

*Model II did (Fig. 1). Model I assumed that plant structural residues were the only POC source, whereas Model II assumed that heavy-POC could be from both plant and microbial residues. Thus, dissolved C can be transformed to heavy-POC via microbial metabolism in Model II.*"

Comment 20:     Would be nice to see the C sequestration projections scaled to ecosystem scale (i.e. MgC/ha/yr) with and without microbial POC mechanisms.

Response: Revised as suggested. In the revision, we have scaled the projections using the measured soil bulk density from the 10 soils. After the scaling, the projected SOC pools at the equilibrium are shown in Mg C ha$^{-1}$ (revised Fig. 4).

Discussion

Comment 21:     Again, you discuss heavy POC, but did not separate this fraction. You show that MBC is correlated to Glucose C, but this explains only a small portion of the glucose C in POC. Is it possible that glucose C is somehow directly sticking to POC and not passing through microbes?

Response: We agree that dissolved C entering into POC cannot entirely attribute to microbial assimilation, as C input can get into soil C pool via other paths, such as direct interactions between plant compounds and mineral surfaces, as well as microbial extracellular decomposition (Craig et al., 2022). Based on Fig. 3 in the manuscript, the behavior of dissolved C entering POC pool is regulated by both physical and biochemical process. In the revision, we have added more discussion about multiple pathways on C sequestration (line 235-241): "*Linear regression analyses indicate that glucose C can enter the POC pool via multiple pathways (Craig et al., 2022). Specifically, glucose-derived POC is positively correlated with the glucose-derived MBC (Fig. 3a), suggesting that the transformation of glucose to POC could be dependent on the microbe-mediated biochemical pathway. Meanwhile, glucose-derived POC is positively correlated with the fraction of clay and silt as well ($R^2$ = 0.92, Fig. 3b), further indicating that dissolved C entering into POC is an interaction of physical and biochemical processes.*"

Comment 22:     These are significant findings that provide evidence that physical transfer pathway and microbial DOC pathway are not distinct. Microbial contributions to POC in these grasslands are significant, I'm interested to know if it is the result of microbial biofilms promoting heavy POC stabilization or possibly fungal bodies larger than 53um? This research opens the door to more exciting research, good work!

Response: We appreciate that the reviewer finds our work scientifically interesting. We are also grateful that the reviewer proposed an important and interesting question on the effects of microbial biofilms and fungal body size on heavy POC stabilization. While the current study was not designed to answer the question, it is definitely worth exploring as a following-up work.

**Technical corrections:**

Comment 23: L31: Consider changing 'As opposed to this' to 'In contrast' -just a suggestion, take it or leave it.

Response: Revised as suggested (line 35-37): "*In contrast, POC is usually considered the product of physically fragmented structural residues and is more susceptible to external environmental changes.*"

Comment 24: L33: Instead of 'roughly' use physically. -just a suggestion, take it or leave it.

Response: Revised as suggested (line 37-38): "*Although physically dividing SOC into POC and MAOC is relatively easy.*"

Comment 25: L34: Remove 'to operate' -just a suggestion, take it or leave it.

Response: Revised as suggested (line 37-38): "*Although physically dividing SOC into POC and MAOC is relatively easy.*"

Comment 26: L42: low molecular weight is more accurate than small-molecular.

Response: Revised as suggested (line 51-54): "*Most of the literature emphasizes that labile plant substrates with low molecular weight – such as glucose and other dissolved C – are primary sources of MAOC through physical absorption and microbial in vivo turnover via cell uptake-biosynthesis-growth-death.*"

Comment 27: L45: sentence is awkward consider changing. i.e. "However, the potential for microbial products derived from labile C to stick to semi-decomposed plant residues and connect with minerals to become POC has received much less attention."

Response: Revised as suggested (line 55-56).

Comment 28:    L48: Remove 'the' in 'the POC' both instances.

Response: Revised as suggested in the revision.

Comment 29:    L87: rather than 'the other' – and the >53 um fraction was considered POC

Response: Revised as suggested (line 116-117): "*The C from less than 53 μm fraction was considered MAOC, and the >53 um fraction was considered POC.*"

Comment 30:    L140: remove duplicate 'effects of'

Response: Revised as suggested (line 181).

Comment 31:    L172: efficiency to efficient.

Response: This sentence has been removed in the revision. We appreciate you pointing out the grammatical mistakes in our manuscript.

RC2: 'Comment on egusphere-2023-1483', Anonymous Referee #2, 27 Sep 2023

Comment 1: Si et al. present data from a laboratory study in which they incubated different soils with $^{13}$C-labeled glucose and measured the label after the end of the incubation in microbial biomass, particulate organic matter (POM), and mineral-associated organic matter (MAOM). The authors find fluxes of C from the glucose to microbial biomass and subsequently to POM and MAOM. The results indicate that flows from dissolved organic C to POM may be relevant to soil C sequestration. The study is timely and provides interesting insights into flows of C from DOM to POM and MAOM. However, I believe that communication of the results could be improved. The authors write at various locations that dissolved compounds form particulate organic C (POC), which connotes that this POC or POM is formed de novo. This is, however, not supported by the data, and it is much more likely that microbes that metabolize the labeled glucose and colonize POM are responsible for the $^{13}$C recovered in that fraction. Moreover, data (e.g., on total POM/POC before and after the incubation) that could substantiate the authors' claims are not provided. I thus suggest that the authors revise their title, abstract, discussion, and conclusions and refrain from using the term "formation" and rather refer to flows of added C to POM (or MAOM). Moreover, the grammar throughout the manuscript should be checked and errors corrected.

Response: We appreciate the reviewer's constructive comments, which have been extremely helpful for us to improve the manuscript. We have revised the manuscript thoroughly. Please find the point-by-point responses below. We hope you will be satisfied with our response.

Specific comments below.

Comment 2: L28 I suggest not referring to stabilization when writing about POC.

Response: We agree that it might not appropriate to refer to POC stabilization. In the revision, we have revised the text in line 27 and other locations. The sentence is in lines 29-30 in the revision: "*Carbon from root exudates can be stabilized in the form of mineral-associated organic C (MAOC), while plant residues can enter the soil as particulate organic C (POC), which has different features from MAOC.*"

Comment 3: L33-25 grammar broken

Response: The sentences has been revised (line 35-40): "*In contrast, POC is usually*

*considered the product of physically fragmented structural residues and is more susceptible to external environmental changes (Benbi et al., 2014; Lugato et al., 2021). Although physically dividing SOC into POC and MAOC is relatively easy, the microbe-mediated SOC dynamics is a continuous process, and it is difficult to separate its biochemical and physical processes completely.*"

Comment 4: L37ff It remains unclear why heavy POC can be a precursor for MAOM formation. I suggest stating the potential formation pathways of MAOC in the text above, for clarity.

Response: We've added more information about the potential formation pathways of MAOC from heavy POC in the revision (lines 42-45): "*Heavy-POC is a complex rich in plant residues, microbial products, and soil minerals. With the gradual decomposition of plant residues in the complex center, heavy-POC gradually fragmented as well, becoming a precursor of MAOC.*"

Comment 5: L40ff I believe the studies cited refer to rhizodeposits and not to root exudates (not sure if Cotrufo et al., 2013 thematize above- or belowground inputs at all). The statement here should be revised accordingly. It could also be mentioned here that root exudates can destabilize C as well (e.g., Keiluweit et al., 2015, Nat. Geosci.).

Response: We appreciate the detailed comment. The sentence has been revised in the revision (lines 47-51): "*Dissolved C input from living root and the rhizodeposits, which has a dominant effect on the net formation of SOC, is considered approximately 2 to 13 times more efficient than litter inputs in forming SOC (Sokol et al., 2019). The Microbial Efficiency-Matrix Stabilization (MEMS) framework also suggests that labile plant C inputs are a major source of microbial products, which are more efficiently utilized by microorganisms than recalcitrant ones (Cotrufo et al., 2013). However, the labile C input also plays a critical role in destabilizing SOC as well (Kuzyakov et al., 2000; Keiluweit et al., 2015).*"

Comment 6: L59ff What was the rational behind sampling these sites? Did site/management have an influence on C flows?

Response: The sampling sites were chosen depending on the major grassland types in Northern China. The chosen sites are broadly representative of meadow grasslands (HL), typical grasslands (DL, GY, XL), and desert grasslands (XH), respectively.

Site and management do have influence on C flows. In the revision, we have added new analyses on the effect of site and management on soil C flows. We have added new results (lines 194-198): "*For the new C sequestration, sites had significant effects on the sequestration of each C pool and respiration, in which glucose-derived MAOC and POC at HL site was significantly higher than that at other sites (Table S3, Fig. S3). Fencing also significantly affected the amount of glucose C entering MAOC as well as the cumulative soil respiration, in which fencing soils show a lower amount of MAOC sequestration and higher soil respiration (Table S3, Fig. S3, S4).*"

We have also added more discussion on the effect of site and management on soil C flows (lines 288-290): "*Meanwhile, higher cumulative respiration in fencing soils can be explained by initial SOC and soil texture, presenting a positive effect of higher SOC content but negative effect of clay and silt content (Table S4).*"

Comment 7: L69ff How was glucose added and how did the authors assure that it was uniformly distributed in the soil?

Response: Before the incubation, 0.5 g of glucose was dissolved in 50 ml of water to make a 10 mg/ml glucose solution. At day 0, 2 ml of glucose solution was slowly dripped into the soil using a pipette gun to keep the solution as uniformly distributed in the soil as possible. Correspondingly, 2 ml of water was added to the control. We have added the information in the revision (lines 96-100): "*After a 7-day pre-incubation, $^{13}C$-labeled glucose (99 atom% $^{13}C$, Shanghai Engineering Research Center of Stable Isotope) was added at a dose of 0.4 mg C $g^{-1}$ soil. The glucose solution was prepared by dissolving 0.5 g of glucose in 50 ml of water to make a 10 mg ml-1 solution. Further, 2 ml of glucose solution was slowly dripped into the soil using a pipette gun to keep the solution as uniformly distributed in the soil as possible. Correspondingly, 2 ml of water was added to the control.*"

Comment 8: L85ff Reference for the methods?

Response: References added in Materials and Methods:

Line 105: "*The chloroform-fumigation-extraction method was used to determine DOC and MBC contents (Vance et al., 1987).*"

Line 112-115: "*The POC and MAOC content were assessed through the particle size fractionation method, which separates SOC into these two pools. Soil samples (10g) were shaken with 30 mL of sodium hexametaphosphate solution (NaHMP, 50 g $L^{-1}$) at 200 rpm.*"

*After 18h, samples were washed with deionized water over a 53 μm sieve in a vibratory shaker (AS 200 control, Retch, Germany)(Sokol et al., 2019)."*

Line 121-122: *"The atom% of MBC in control and treated soils was determined using a two-pool mixing model (Fang et al., 2018)."*

Comment 9: L132ff How were the models validated?

Response: We appreciate the insightful comment. In the revision, the model structure has been modified following the first reviewer's comments. The model was calibrated and validated using the $CO_2$ emission rates and pool sizes at the end of incubation (Fig. S4). The $CO_2$ emission data were divided into two groups: 7 out of the 9 flux measurements for each soil were randomly selected for the model calibration, while the other 2 measurements were used for the model validation (revised Fig. S5). We have added the information in the Materials and Methods section (line 164-167): *"The models were calibrated using soil C pools and $CO_2$ emission rate data through the adaptive Metropolis algorithm (Haario et al., 2001; Hararuk et al., 2014). The $CO_2$ emission data were divided into two groups: 7 out of the 9 flux measurements for each soil were randomly selected for the model calibration, while the other 2 measurements were used for the model validation."*

[Figure]

**Revised Figure S5. Model validation using soil respiration data.** Two randomly selected measurements from $CO_2$ emission data per site were used for model validation. Diagonal black dotted line is 1:1 line. Please see Table 1 for the abbreviations.

Comment 10:     L149/150 What proportion of initially added glucose is this?

Response: 84.28 –175.80 mg kg$^{-1}$ soil of the glucose C is equivalent to 21.07% – 43.95% of the initial glucose addition. For convenience, we added a new y-axis in revised Fig. 2 to show the relative contribution of glucose C to different C pools. Meanwhile, we added more description to revision in lines 200-203: "*Across the 10 soils, 84.28 –175.80 mg kg$^{-1}$ soil of the glucose C (equivalent to 21.07% – 43.95% of the initial glucose addition) retained in the soil after the 102-day incubation, among which 1.58% – 28.00%, 48.73% – 75.51%, 20.34% – 35.80% of retained glucose C distributed in POC, MAOC and MBC, respectively. At the end of incubation, the proportion of total POC that is from glucose C was 0.16% – 0.67%.*"

Comment 11:     L151 glucose-derived POC and MAOC were correlated, not dependent.

Response: The sentence has been revised as suggested (line 205-206): "*glucose-derived MAOC and POC were correlated with glucose-derived MBC.*"

Comment 12:     L155 Compared to what did the model under-/overestimate turnover?

Response: The parameters were compared between Model I and Model II. In the revision, we have made it clearer in line 212-213: "*On average, compared to Model II, Model I showed greater $k_L$, $k_M$, $f_{BD}$, $f_{MB}$, $f_{DM}$, but smaller $k_D$, $k_B$, $k_H$, $f_{DL}$, $f_{DH}$, $f_{MH}$.*"

Comment 13:     L170ff I believe that this statement is erroneous, i.e., the correlation of glucose-derived POC with glucose-derived MBC indicates that microbial processing/metabolization of DOM and subsequent colonization of POM by these microorganisms explains the observed pattern.
That is, DOM or microorganisms do not form POM de novo but "attach" to existing POM. So, what the authors observe in their study is basically the decomposition process and no de-novo formation of POC/POM. The authors could use data on total POC/POM values in their soils before and after the incubation to substantiate their claims, but such data are absent. As such, the title of the manuscript is misleading as well. I thus suggest that the authors frame their discussion differently and refrain from using the term POC formation, which connotes that DOC forms POC de novo, but only refer to flows of C to POC. The abstract and conclusions should be adapted accordingly.

Response: We appreciate the valuable comment. We agree with the reviewer that the results

in the current work provide direct evidence of C flows from DOC to POC instead of POC formation de novo. The reviewer suggests using data on total POC/POM values in the soils before and after the incubation to test the POC formation de novo. Although we unfortunately did not conduct the fractionation before the incubation, we do value the reviewer's suggestion. We compared POC and MAOC contents in the control and glucose addition treatments. Compared to the total MAOC or POC, the amount of glucose-induced change was tiny, which was within the range of error bar (Fig. R1). The new analysis does not support the claim that the flow of C to POC is the POC formation de novo. Therefore, we have revised the title, Abstract, Introduction, Results and Discussion according to the reviewer's suggestions. In the revision, we have replaced POC formation with dissolved C flow to POC. The revised title is "*Dissolved carbon flow to particulate organic carbon enhances soil carbon sequestration*".

[Figure]

**Figure R1. MAOC and POC contents of glucose-added and control treatments at the end of incubation.** The top half of the figure represents MAOC, and the bottom half represents POC. filled bars represent the glucose-added treatment and open bars represent the control. Mean ± SD.

Comment 14:   L172 The study by Sokol does not support the claims by the authors since inputs by living roots encompass structural compounds, e.g., sloughed-off cells, which can be a substantial contributor to POM, and not just exudates.

Response: We appreciate the reviewer pointing out our problem. In the revision, we have removed the statement.

Comment 15:     L172 The authors use the term POC formation, which I believe is misleading since they only show POC derived from glucose but no overall POC or POM values that would substantiate additional formation of POC.

Response: We agree with the reviewer. As responded above, we have revised the title, Abstract, Introduction, Results and Discussion according to the reviewer's suggestions. In the revision, we have replaced POC formation with dissolved C flow to POC. The revised title is "*Dissolved carbon flow to particulate organic carbon enhances soil carbon sequestration*".

Comment 16:     L174 "was positively correlated"; this correlation could indicate that the higher the clay and silt content, the more aggregates and the more POM is protected from decomposition. In the following lines, the authors indirectly refer to the aggregation process, which could be explicitly referred to as such.

Response: We revised "positively dependent on" to "positively correlated with" (line 208 in the revision). Additionally, we have directly referred to the aggregation process in the line 246-247: "*Meanwhile, the higher clay and silt content means the more microaggregates and more POC protected from decomposition (Wang et al., 2003).*"

Reference:

[revised manuscript text omitted]

---

## Author Response (AR2)

**Letter of Responses**

Author Note: Comments from editors and reviewers are in black and the responses follow in blue. Line numbers in the responses are those in the marked-up revision. Reviewers' comments have been numbered for the easy reading.

Dear Dr. Liang,

Thank you for your careful response to the original reviewers (original R1 & R2) which are appreciated by current R2 and the editorial team (original R1 was not available to reply). As you can see, R3 (current R1) has presented new and valuable comments on your revised manuscript. Could you please now provide a response to R3?

Best regards,
Kate Buckeridge

Response: Dear Dr. Buckeridge, we greatly appreciate the constructive comments from the editor and three knowledgeable reviewers. We are also grateful to you for offering the opportunity to us to revise the manuscript. Detailed point-by-point responses to comments of reviewers are below. We hope you will find our revision satisfactory.

Report #2
Anonymous referee #1:
Comment: I want to thank the authors for their thorough and comprehensive revision of this manuscript. Excellent work! I recommend to accept as is.

Response: We appreciate the reviewer finds our revision satisfactory. We are grateful for the reviewer's constructive comments during the peer-review process.

Report #1
Anonymous referee #3:
**Overall Comments:**

Note that I am newly reviewing this paper after the first round of revisions. This paper uses isotopically labeled glucose additions to investigate POM and MAOM formation from dissolved C. It finds evidence for dissolved C contributing to POM, in opposition to the widely cited and evidenced two-pathway model (Cotrufo et al., 2015). This is an exciting finding and I think the topic of this paper is very timely and addresses an important uncertainty in the SOM

literature. However, I felt some of the methodological choices require greater justification and the presentation of the results could be more clear:

Response: We appreciate the reviewer's recognition and constructive comments on our manuscript. Based on the reviewer's suggestions, we have revised the manuscript thoroughly. Please find the point-by-point responses below. We hope that the reviewer will find our revision satisfactory.

Comment 1: Particularly, it was unclear to me why the authors focus on heavy POM but do not carry out a fractionation scheme that provides a heavy POM pool. I see this is addressed in the response to reviewers but it is important that the authors make it clear why they did this in the methods.

Response: We appreciate the reviewer's suggestion to add justification for our choice in the Methods. In the revision, we have added more details about why we isolated heavy-POC in the modeling analysis (lines 143-147) but not in the incubation experiment (lines 117-122):

Line 117-122: "*The experimental design was a trade-off between the incubator's space (which determines the jar volume and soil samples in jars) and the total number of jars. With the limited samples, we decided to do the size separation (i.e., POC vs. MAOC) instead of the density fractionation because we anticipated that the size separation might provide more insights into SOC dynamics and is more related to microbial processes according to the literature (Lavallee et al., 2020). Despite that, both light-POC and heavy-POC were included in the following modeling analysis to broaden the implication of the experiment.*"

Line 143-147: "*Models designed based on the data of soil C pools and $CO_2$ emission fluxes in the incubation experiment. Here, it is important to note that in both soil C models, POC is divided into two pools: the light-POC and the heavy-POC. This is because heavy-POC is a plant residue-microbial product-soil mineral complex, which is more likely to be the destination of dissolved C inputs than light-POC, which only comprises plant residues (Samson et al., 2020). Therefore, the two POC pools were modeled separately.*"

Comment 2: Additionally, I am confused why the authors carry out DOC and DOC+POC addition experiments in their model – it is unclear what question this is addressing. This requires justification so it is more clear what the reader is supposed to learn from this work.

Response: We set up those two C input scenarios based on the incubation experiment as well as the natural C input conditions. Firstly, the "DOC input only" scenario was set to be consistent with our incubation experiment, which only included dissolved C input (i.e., glucose). In the absence of structural C inputs, newly formed POC only originates from dissolved C input, which allows us to determine directly how the process of dissolved C flow to POC impacts C sequestration. Secondly, the "DOC+POC input" scenario was set up to mimic the natural C input in the field. This scenario illustrates that even with structural C inputs, the process of dissolved C flow to POC can still have a significant impact on C sequestration. In the revision, we added more description of these two C input scenarios:

Line 180-183: "*For each model, we set up two C input scenarios. To fit with the incubation experiment including dissolved C input only, we set up the scenario of "DOC input only." To make the prediction closer to the natural C input in the field, we set up the scenario of "DOC+POC input."*"

Comment 3: In the title, abstract, discussion, and conclusion, there is large emphasis on the impacts of long-term C sequestration but is unclear if the study evaluates long-term C sequestration. Without this evaluation, I don't believe this claim is supported and it would be useful to use more cautious language and consider that sequestration implies storage over a long period, rather than solely new formation of soil C. Throughout the paper, C sequestration can often be replaced with C pools or stocks for concentrations and area-based estimates, respectively.

Response: We appreciate reviewer's suggestion to be careful in using the term *C Sequestration*. We agree that C sequestration is a long-term process. We are aware that results from the incubation experiment may not be appropriate to demonstrate the change in C sequestration. That is one of the reasons we used the model analysis, in which we ran the model to the equilibrium (i.e., the steady state after 500 years).

We do value the reviewer's suggestion. In the revision, we have revised thoroughly to avoid C sequestration when describing the results of incubation experiment and replace it with new C formation. In the modeling analysis, we hope the reviewer would agree that using C sequestration is appropriate to describe the long-term SOC simulation. Adding the process of dissolved C flow to POC leads to a net increase in the predicted SOC stock.

Comment 4: Finally, the main claim of the work, that dissolved C contributes significantly to

POC, seems to be undermined by the result that added DOC represented less than 1% of total POC pool – this should be addressed more clearly to ensure the conclusions in the paper hold true to the data.

Response: Glucose-derived POC is indeed only a small part of the total POC pool in the incubation experiment. because the experiment was designed based on the annual C input relative to soil microbial biomass in the studied sites, which is a widely used methodology (e.g.,Ghee et al. (2013); Zhang et al. (2016)). Compared with the initial SOC, glucose is added in a small amount, 0.4 mg C g$^{-1}$ soil, which only accounts for 0.67% ~ 5.70% of the initial SOC content (varying with sites). Even though glucose has a great transformation rate, the C sequestration amount is relatively small in the experiment. Similarly, when we focus on the amount of glucose derived-MAOC as a percentage of total MAOC (0.26% ~ 1.46%), this is also a small value because it is constrained by the limited C input. Therefore, we consider that newly formed C accounts for only a small fraction of the total POC is reasonable.

Compared to the ratio of glucose-derived POC to total C pool, we focus more on the proportion of glucose derived-POC to total glucose-C, because it fully represents the distribution of the newly added C between the different C pools. This proportion is relatively larger and more pronounced, i.e., 1.58% ~ 28.00% of the dissolved C input enters the POC pool. Therefore, we consider that this data is sufficient to support our conclusion that dissolved C contributes significantly to POC.

**Specific Comments:**

Comment 5: Line 37: I believe the authors mean "coarse MAOC", not "course MAOC".

Response: We appreciate the reviewer pointing out spelling errors. We have revised it as suggested.

Comment 6: Lines 43-45: This sentence is unclear. I think the authors mean to say that heavy POC may be a precursor to MAOC, given heavy POC is a combination of plant residues, microbial products, and soil minerals?

Response: Yes, our intent is consistent with the reviewer's understanding. We have rewritten the sentence to more clearly describe the process from heavy-POC to MAOC in the revision (lines 35-41).

Line 35-41: "*During the gradual decomposition of plant residues by microorganisms, the surface of fresh litter combines with soil minerals in the presence of microbial products (which act as binders in the soil), forming heavy-POC (or coarse-MAOC, >53 μm and >1.6 – 1.85 g cm$^{-3}$) (Samson et al., 2020). The plant-soil interface in heavy-POC promotes the formation of soil aggregates directly, becoming the hotspot for SOC formation (Witzgall et al., 2021). Meanwhile, given that heavy-POC is a combination of plant residues, microbial products, and soil minerals, and that heavy-POC has a similar C:N ratio to MAOC, it is reasonable to hypothesize that the fragmentation of heavy-POC promotes the formation of MAOC and is the precursor for MAOC (Samson et al., 2020; Zhang et al., 2021).*"

Comment 7: Line 46: Note that MEMS 2.0 no longer contains this pool so I am not sure this line should remain in the manuscript. See Zhang et al., 2021. Zhang, Y., Lavallee, J.M., Robertson, A.D., Even, R., Ogle, S.M., Paustian, K., Cotrufo, M.F., 2021. Simulating measurable ecosystem carbon and nitrogen dynamics with the mechanistically defined MEMS 2.0 model. Biogeosciences 18, 3147-3171.

Response: We appreciate the reviewer for reminding us that the heavy-POC pool is no longer included in MEMS 2.0. We have removed the sentence as the background of our research in the revision.

Comment 8: Line 47: Dissolved C input from living roots are included as rhizodeposits right?

Response: We agree with the reviewer that dissolved C input from living roots is included as rhizodeposits. We have revised the text in lines 48-49: "*Dissolved C input from living roots (i.e. rhizodeposits), which has a dominant effect on the net formation of SOC, is considered approximately 2 to 13 times more efficient than litter inputs in forming SOC.*"

Comment 9: Line 70 and throughout: I believe this should be "fenced and grazed grasslands" to be grammatically correct?

Response: Revised as suggested. In the revision, we replaced "fencing and grazing grasslands" with "fenced and grazed grasslands".

Comment 10:  Line 83: I am not following this last hypothesis – increased microbial C use efficiency from microbial use of DOC that then becomes POC or of POC that becomes MAOC?

Response: The final hypothesis has been modified to focus more on modeling analysis results instead of microbial C use efficiency, which would lead to a clearer and less ambiguous statement (lines 79-80). We hope this will help the reader better understand our hypothesis.

Line 79-80: "*Finally, neglecting the process of dissolved C flow to POC leads to an underestimation of SOC sequestration.*"

Comment 11: Lines 112 and 135: I am confused by the authors use of a size fractionation when they seem to be interested in heavy POC dynamics, given the modeling. As noted above, there should be justification for this choice in the methods.

Response: We appreciate the reviewer's suggestion to add justification for our choice in the Methods. As response to Comment 1 above, we have added the justification in the revision.

Comment 12: Line 164: Were the two modeled POC pools summed together for calibration?

Response: Yes, since we only measured the size of total POC, heavy-POC pool and light-POC pool were summed together for the calibration.

Comment 13: Line 173: It is unclear why the authors choose these 100 parameter sets for further work. It is my understanding that the calibration step explained above this line provides the parameter values.

Response: In the modeling experiments, we randomly choose 100 sets of parameters is to reduce the cost of computation. During the calibration, we set up 30,000 times of simulations. Based on the reception rates at different sites, the model ended up receiving about 9,000 to 15,000 sets of parameters. In the later simulation experiments we run the model for 500 years until the steady state, but using all parameter sets would be time-consuming. Therefore, we randomly select 100 sets of parameters in the subsequent predictions. Randomly selecting a subset of parameters is proven to be representative to reproduce the original parameter distributions and also improves the simulation efficiency (Xu et al., 2006; Liang et al., 2018).

Comment 14: Line 174: It is unclear why there are two input scenarios – what question is this testing?

Response: We appreciate the insightful comment. The two scenarios we set were to test

how the process of dissolved C flow to POC affects soil C sequestration under the both experimental and field condition. In the revision, we added more description of these two C input scenarios. Please see our detailed response in Comment 2 above.

Comment 15:     Line 178 and throughout: As noted above, it is unclear how the authors are testing "long-term SOC sequestration". Are the models being run forward for many years? If so, how much trust can we put in models that are so specifically parameterized? I think the models in this work can help us understand mechanisms but given parameterization was done for individual sites and treatments, the models are not very generalizable. Please consider the definition of sequestration in Don et al., 2023 when using this term. Don, A., Seidel, F., Leifeld, J., Kätterer, T., Martin, M., Pellerin, S., Emde, D., Seitz, D., Chenu, C., 2023. Carbon sequestration in soils and climate change mitigation—Definitions and pitfalls. Global Change Biology, e16983.

Response: Based on Don et al. (2023), we have carefully re-examined the use of *C sequestration* in the manuscript. In the modeling experiment, models were run for 500 years to reach a steady state after calibration. The results show a net increase in the SOC prediction after adding the process of dissolved C flow to POC, which is consistent with the definition of C sequestration. Meanwhile, we agree that our models are too specifically parameterized to be generalizable. However, given that the data from all 10 sites had similar results in the model, we hope the reviewer would agree that it is reasonable to state that it can help us understand mechanisms of the dissolved C flow to POC on C sequestration.

Comment 16:     Line 203: As noted above, less than a percent is a small contribution – does this not undermine the conclusions that DOC flows to POC and contributes to C sequestration?

Response: We appreciate the constructive comment. Please see our response to Comment 4.

Comment 17:     Line 209: I recommend creating a modeling results section and making the above section (3.2) an incubation results section.

Response: We appreciate the reviewer's suggestion on creating a new section for the modeling results. In the revision, we moved the results of modeling analysis to Section 3.3, *modeling analysis of soil C dynamics*.

**Comment 18:** Line 213: The absence here is the comparison of Model 1 to Model 2, correct? Please clarify.

Response: Yes, the absence of $f_{HB}$ here is the comparison of Model I to Model II. To clarify, we rewrite the original sentence in revision (line 226): "*Compared with Model II, the absence of the $f_{HB}$ in Model I affects other parameters differently.*"

**Comment 19:** Line 234: Is this 36% of the glucose C in POC and MAOC? Line 203 suggests a much smaller amount of the total C pool is glucose-derived POC. Also, I would say this is POC and MAOC formation – sequestration requires evaluating POC and MAOC over time (again see Don et al., 2023).

Response: In the previous marked-up revision, 36.49% in the Line 234 was calculated through $\frac{glucose\ derived-POC}{glucose\ derived-POC\ +\ glucose\ derived-MAOC}$. It is not same with the proportion in the Line 203, with is calculated through $\frac{glucose\ derived-POC}{total\ POC}$. We were attempt to show the proportion of 36.49% to illustrate that the pathway from DOC to POC accounted for a significant proportion of the SOC sequestration. In the revision, to make the statement clearer, we have removed this sentence.

Meanwhile, we agree that changes in the C pool observed in short-term incubation experiment belongs to C formation rather than C sequestration. In the revision, we have revised thoroughly to avoid using the term C sequestration to describe the results of incubation experiment.

**Comment 20:** Line 262: I would say "potential usefulness" rather than "necessity". I don't think one experiment justifies adding this process to SOC dynamic models.

Response: Revised as suggested (line 273).

**Comment 21:** Figure 4: Please make the outliers for each boxplot more visible, either by outlining in black or making them black dots.

Response: In the revision, we have marked the outliers in Figure 4 and Figure S8 in black, hoping to make it clearer to the reader.

[Figure]

**Revised Figure 4: Modeled SOC content at steady state under two types of C input conditions.** The two different C input scenarios for each site are separated by a dotted line. The upper and lower ends of boxes denote the 0.25 and 0.75 percentiles, respectively. The solid line and dot in the box mark the median and mean of each dataset. The open circles denote outliers. Asterisks represent significant differences between Model I and Model II (*$P < 0.05$, **$P < 0.01$, ***$P < 0.001$).

**Reference:**

Don, A., Seidel, F., Leifeld, J., Kätterer, T., Martin, M., Pellerin, S., Emde, D., Seitz, D., and Chenu, C.: Carbon sequestration in soils and climate change mitigation—Definitions and pitfalls, Glob. Change Biol., 30, https://doi.org/10.1111/gcb.16983, 2023.

Ghee, C., Neilson, R., Hallett, P. D., Robinson, D., and Paterson, E.: Priming of soil organic matter mineralisation is intrinsically insensitive to temperature, Soil Biology and Biochemistry, 66, 20-28, https://doi.org/10.1016/j.soilbio.2013.06.020, 2013.

Lavallee, J. M., Soong, J. L., and Cotrufo, M. F.: Conceptualizing soil organic matter into particulate and mineral-associated forms to address global change in the 21st century, Glob. Change Biol., 26, 261-273, https://doi.org/10.1111/gcb.14859, 2020.

Liang, J., Zhou, Z., Huo, C., Shi, Z., Cole, J. R., Huang, L., Konstantinidis, K. T., Li, X., Liu, B., Luo, Z., Penton, C. R., Schuur, E. A. G., Tiedje, J. M., Wang, Y. P., Wu, L., Xia, J., Zhou, J., and Luo, Y.: More replenishment than priming loss of soil organic carbon with additional carbon input, Nature Communications, 9, 3175, https://doi.org/10.1038/s41467-018-05667-7, 2018.

Samson, M.-É., Chantigny, M. H., Vanasse, A., Menasseri-Aubry, S., and Angers, D. A.: Coarse mineral-associated organic matter is a pivotal fraction for SOM formation and is sensitive to the quality of organic inputs, Soil Biology and Biochemistry, 149, https://doi.org/10.1016/j.soilbio.2020.107935, 2020.

Witzgall, K., Vidal, A., Schubert, D. I., Hoschen, C., Schweizer, S. A., Buegger, F., Pouteau, V., Chenu, C., and Mueller, C. W.: Particulate organic matter as a functional soil component for persistent soil organic carbon, Nature Communications, 12, https://doi.org/10.1038/s41467-021-24192-8, 2021.

Xu, T., White, L., Hui, D. F., and Luo, Y. Q.: Probabilistic inversion of a terrestrial ecosystem model: Analysis of uncertainty in parameter estimation and model prediction, Global Biogeochemical Cycles, 20, https://doi.org/10.1029/2005gb002468, 2006.

Zhang, H., Ding, W., Luo, J., Bolan, N., Yu, H., and Zhu, J.: Temporal responses of microorganisms and native organic carbon mineralization to 13C-glucose addition in a sandy loam soil with long-term fertilization, European Journal of Soil Biology, 74, 16-22, https://doi.org/10.1016/j.ejsobi.2016.02.007, 2016.

Zhang, Y., Lavallee, J. M., Robertson, A. D., Even, R., Ogle, S. M., Paustian, K., and Cotrufo, M. F.: Simulating measurable ecosystem carbon and nitrogen dynamics with the mechanistically defined MEMS 2.0 model, Biogeosciences, 18, 3147-3171, https://doi.org/10.5194/bg-18-3147-2021, 2021.

---

## Author Response (AR3)

**Letter of Responses**

Author Note: Comments from editors and reviewers are in black and the responses follow in blue. Line numbers in the responses are those in the marked-up revision.

Dear Dr Liang,

R1 (original R3) has submitted further comments; could you please respond to this review?

Best regards,
Kate Buckeridge

Response: Dear Dr. Buckeridge, we are grateful for the follow-up comments from the reviewer. We have revised the manuscript based on the new comments. Detailed point-by-point responses to the comments are below. We hope you will find our revision satisfactory.

Report #1
Anonymous Referee #3:
I appreciate the authors addressing my comments and believe the manuscript will be of interest to readers of SOIL. As stated in my previous review, I think this work addresses an important uncertainty in the SOM field.

Response: We appreciate the reviewer's recognition of our work. We are grateful for the reviewer's constructive comments during the peer-review process.

I have a few minor follow-up comments:

In response to my comment 3: I appreciated this response and think it would be helpful to a reader to state that running the model to steady state was a 500-year model run. That was not clear to me and makes the use of "C sequestration" more valid.

Response: We agree that adding information about "500-year model run" would help readers understand the model's steady state and long-term predictions of C sequestration. We have revised as suggested (line 178-179): "*The calibrated models were run for 500 years to steady states to compare the modeled SOC change under different scenarios.*"

In response to my comment 4: I suggest that the authors add the amount of glucose-derived MAOC as a percentage of total MAOC at line 212 so the reader can compare POC and

MAOC. Having that information from the authors was helpful for me to understand that the contribution to both POC and MAOC overall was small. Additionally, I think it is important for the authors to acknowledge that DOC still contributed dominantly to MAOC, and so a complete inconsistency with the two-pathway model is not fully supported (lines 261-264).

Response: We agree that adding more information about MAOC is helpful to readers comparing the POC and MAOC formation. We have revised the text in lines 204-205: "*At the end of incubation, the proportion of glucose-derived C to total POC was 0.16% ~ 0.67% and to total MAOC was 0.26% ~ 1.46%.*"

Additionally, we agree that our results are not completely inconsistent with the two-pathway model. They are more like a complement to the framework. To express more accurately, we have revised the text in line 253: "*The result that labile C can enter the POC pool is partially inconsistent with the two-pathway framework.*"

In response to my comment 12: It would be helpful to say in the methods in the manuscript that the authors calibrated to the sum of heavy and light POC.

Response: In the revision, we have added more descriptions about POC calibration (line 165-166): "*For the POC pool, heavy-POC pool and light-POC pool were summed together for the calibration.*"

In response to my comment 13: To clarify, the 100 parameter sets used in the subsequent predictions were chosen from the posterior distribution of the Metropolis algorithm? That would be helpful to state to readers.

Response: Yes, the 100 parameter sets used in the SOC predictions were chosen from the posterior distribution of the Metropolis algorithm. We have added this statement in the revision (line 174-175): "*After the model calibration and validation, we randomly selected 100 sets of parameters from posterior PDFs of the adaptive Metropolis algorithm for further modeling experiments.*"